# A de novo evolved gene contributes to rice grain shape difference between *indica* and *japonica*

Rujia Chen [1,2,9], Ning Xiao[3,9], Yue Lu[1,2,9], Tianyun Tao[1], Qianfeng Huang[1], Shuting Wang[1], Zhichao Wang[1], Mingli Chuan[1], Qing Bu[1], Zhou Lu[1], Hanyao Wang[1], Yanze Su[1], Yi Ji[1], Jianheng Ding[1], Ahmed Gharib [1,4], Huixin Liu[1,2], Yong Zhou[1,2], Shuzhu Tang[1,2], Guohua Liang[1,2], Honggen Zhang[1,2], Chuandeng Yi[1,2], Xiaoming Zheng[5], Zhukuan Cheng[1,2], Yang Xu[1,2], Pengcheng Li[1,2], Chenwu Xu [1,2] ✉, Jinling Huang [6,7,8] ✉, Aihong Li [3] ✉ & Zefeng Yang [1,2] ✉

The role of de novo evolved genes from non-coding sequences in regulating morphological differentiation between species/subspecies remains largely unknown. Here, we show that a rice de novo gene *GSE9* contributes to grain shape difference between *indica/xian* and *japonica/geng* varieties. *GSE9* evolves from a previous non-coding region of wild rice *Oryza rufipogon* through the acquisition of start codon. This gene is inherited by most *japonica* varieties, while the original sequence (absence of start codon, *gse9*) is present in majority of *indica* varieties. Knockout of *GSE9* in *japonica* varieties leads to slender grains, whereas introgression to *indica* background results in round grains. Population evolutionary analyses reveal that *gse9* and *GSE9* are derived from wild rice Or-I and Or-III groups, respectively. Our findings uncover that the de novo *GSE9* gene contributes to the genetic and morphological divergence between *indica* and *japonica* subspecies, and provide a target for precise manipulation of rice grain shape.

As a driving force of evolutionary innovation, new genes can evolve from pre-existing genes or gene fragments, including gene duplication, horizontal gene transfer, gene fusion and other mechanisms[1]. By contrast, new protein-coding genes may also originate de novo from previous non-coding sequences[2]. Although de novo genes are derived from 'junk DNA' sequences, recent evidence revealed that they are important for the genomic innovations and biological processes of certain eukaryotes[2–6]. However, the roles of de novo genes in morphological differentiation between species/subspecies remain largely unknown.

Asian cultivated rice (*Oryza sativa* L.) is the staple crop for more than half of the global population[7], and has evolved into several sub-species with significant divergence in genomic and morphological characteristics[8]. Grain shape, a typical complex quantitative trait

[1]Jiangsu Key Laboratory of Crop Genomics and Molecular Breeding/Zhongshan Biological Breeding Laboratory/Key Laboratory of Plant Functional Genomics of the Ministry of Education, Agriculture College of Yangzhou University, Yangzhou 225009, China. [2]Jiangsu Co-Innovation Center for Modern Production Technology of Grain Crops/Jiangsu Key Laboratory of Crop Genetics and Physiology, Yangzhou University, Yangzhou 225009, China. [3]Institute of Agricultural Sciences for Lixiahe Region in Jiangsu, Yangzhou 225009, China. [4]Rice Department, Field Crops Research Institute, ARC, Sakha, Kafr El-Sheikh 33717, Egypt. [5]National Key Facility for Crop Gene Resources and Genetic Improvement, Institute of Crop Sciences, Chinese Academy of Agricultural Sciences, Beijing 100081, China. [6]Department of Biology, East Carolina University, Greenville, NC 27858, USA. [7]State Key Laboratory of Crop Stress Adaptation and Improvement, Key Laboratory of Plant Stress Biology, School of Life Sciences, Henan University, Kaifeng 475004, China. [8]Key Laboratory for Plant Diversity and Biogeography of East Asia, Kunming Institute of Botany, Chinese Academy of Sciences, Kunming 650201, China. [9]These authors contributed equally: Rujia Chen, Ning Xiao, and Yue Lu. ✉e-mail: cwxu@yzu.edu.cn; huangj@ecu.edu; yzlah@126.com; zfyang@yzu.edu.cn

genetically controlled by multiple genes, is a key morphological character distinguishing the two main subspecies *indica/xian* and *japonica/geng*, which generally exhibit slender and ovate grains, respectively. Grain shape is also a key determinant of rice yield, appearance quality and market values[9]. Therefore, understanding the genetic basis of grain shape could shed lights on the evolution of cultivated rice and provide novel strategies for rice molecular breeding.

Numerous quantitative trait loci (QTLs)/genes that control rice grain shape have been identified thus far[10]. For instance, *GRAIN SIZE 3* (*GS3*) is a major QTL for grain size, and natural variation of this gene contributes to the difference of grain length between *indica* and *japonica* varieties[11,12]. *GRAIN SIZE ON CHROMOSOME 5* (*GSE5*) gene that controls grain size diversity in cultivated rice was also identified through a genome-wide association study (GWAS) combined with functional analysis[13]. Nevertheless, whether and how other genes are involved in grain shape differentiation between rice subspecies remain to be explored.

Here, we report a rice de novo gene that contributes to grain shape difference between subspecies. The identification of this gene provides a target for precise genetic manipulation of rice grain shape, which has major agricultural and commercial implications.

## Results

### Identification of the *GSE9* locus for grain shape by GWAS

We performed GWAS analyses to understand the genetic architecture of grain shape using the panel of both *indica* and *japonica* varieties (Supplementary Fig. 1 and Supplementary Data 1). In addition to several well-known loci such as *GS3*[11], *GIF1*[14] and *GSE5/GW5*[13], our analyses identified a locus on chromosome 9 that was responsible for both grain length (GL) and grain width (GW) (Fig. 1a, b), here designated as *GRAIN SHAPE ON CHROMOSOME 9* (*GSE9*). The average decay of linkage disequilibrium (LD) distance was estimated about 110 kb in this population ($r^2 = 0.1$) (Supplementary Fig. 2). The potential candidate genes of *GSE9* locus were further analyzed within the 220-kb interval centered on the leading SNPs (chr09:3105176 for GL; chr09:3134839 for GW), and 15 candidate genes were then identified for *GSE9* locus (Fig. 1c and Supplementary Table 1). We randomly selected four lines from each of slender-grain and round-grain varieties to measure the transcript levels of these 15 genes. We observed that the gene XII (LOC_Os09g06719) had no detectable expression levels in panicles of the selected slender-grain varieties, but relatively high levels of expression in those of round-grain varieties, while the other genes showed no significant difference in expression levels in panicles from the slender-grain and round-grain varieties (Fig. 1d, e and Supplementary Fig. 3). Therefore, the LOC_Os09g06719 gene, encoding an open reading frame (ORF) without annotated domains, was the most likely candidate for *GSE9* locus. We then designated this gene as *GSE9*.

### Functional validation of the role of *GSE9* in regulating grain shape

Given that *GSE9* encodes a protein without known sequence characteristics, the 5' and 3' RACE were used to amplify the full-length cDNA of this gene (Supplementary Fig. 4), and the results revealed that *GSE9* is 900 bp in length and contains an ORF of 324 bp encoding 107 amino acids. To confirm the effect of *GSE9* on grain shape, the CRISPR/Cas9 system was used to specifically disrupt the *GSE9* gene in the *japonica* variety Zhonghua 11 (ZH11) (Fig. 2a–c). ZH11-Cas9 mutants showed no significant differences in main agronomic traits from the wild-type plants, but instead exhibited longer and narrower grains (Fig. 2d–f and Supplementary Table 2). The length of grains was increased by 5.53% and 4.66%, while the grain width reduced by 2.50% and 4.56% in ZH11-Cas9-1 and ZH11-Cas9-2, respectively (Fig. 2f). We also generated two homozygous knockout lines in the *japonica* variety Nipponbare (NIP), and similar phenotypic changes were also observed in the NIP-Cas9

mutants (Supplementary Fig. 5 and Supplementary Table 3). In addition, overexpression of *GSE9* significantly increased 1000-grain weight and grain yield in both the ZH11 and NIP backgrounds, suggestive of its application potential in rice yield breeding (Supplementary Fig. 6). These results revealed that *GSE9* was a regulator of rice grain shape.

RNA in situ hybridization was further performed to detect the localization of *GSE9* transcripts in young panicles in detail (Fig. 2g, h). No signal was detected in the negative control (Fig. 2g). By contrast, *GSE9* hybridization signals were observed in young panicles, including anthers, palea and lemma of spikelet hulls (Fig. 2h). We also generated the transgenic rice plants expressing the β-glucuronidase (GUS) reporter gene driven by the *GSE9* promoter to examine the expression patterns of *GSE9*. The *GSE9pro::GUS* transgenic plants had no obvious phenotypic differences including grain shape compared to the wild-type ZH11 (Supplementary Fig. 7). GUS staining showed that *GSE9* was expressed in all detected tissues, including the stem, sheath, root, leaf, node, spikelet and panicle (Supplementary Fig. 8 and Fig. 2i–k). Strong GUS activity was observed in younger spikelet hulls and panicles (Fig. 2i–k). In line with the GUS staining assays, quantitative reverse transcription-PCR (qRT-PCR) analysis revealed that the transcription level of *GSE9* is relatively high in young panicles of 2–9 cm in length, and decreased as the panicle matures (Fig. 2l), suggestive of a role of *GSE9* in grain development of rice. These findings, together with grain shape changes in *GSE9* knockout mutants, again suggested that *GSE9* (LOC_Os09g06719) was the most likely candidate gene for *GSE9* locus.

### *GSE9* controls rice grain shape by regulating both cell expansion and cell proliferation

Rice grain shape has been proposed to be restricted by the spikelet hull, which is coordinately controlled by cell proliferation and cell expansion[15,16]. To further understand the function of *GSE9* in grain shape, we observed the cell number and cell size in spikelet hulls of ZH11 and *GSE9* overexpression/knockout lines. The changes of spikelet length and width were consistent with grain shape changes in *GSE9* transgenic lines (Supplementary Figs. 6, 9). The spikelet hulls of ZH11-Cas9-1 had fewer epidermal cells in the grain-width direction than those of ZH11 and no significant changes in cell number in the grain-length direction, whereas both cell length and width were increased in ZH11-Cas9-1 spikelet hulls (Fig. 3a–k). By contrast, the ZH11-OE-1 spikelet hulls contained more cells in both the grain-width and grain-length directions, but possessed smaller cells compared to ZH11 (Fig. 3a–k). In addition, we noticed that several cell cycle-related genes were down-regulated in young panicles of ZH11-Cas9-1 mutant and, conversely, up-regulated in ZH11-OE-1 (Fig. 3l, m). By contrast, the transcript levels of cell expansion-related genes were elevated in ZH11-Cas9-1, but suppressed in ZH11-OE-1 (Fig. 3l, n). These findings suggested that *GSE9* functions as a regulator of both cell proliferation and cell expansion. A total of 64 well-documented genes regulating grain shape were also selected for the analysis of expression changes in ZH11-OE-1 *vs* ZH11 and ZH11-Cas9-1 *vs* ZH11 comparisons. The results showed that none of these known grain shape-related genes had a significantly opposite trend of expression change in the two comparisons (Supplementary Data 2), suggesting that the regulation of *GSE9* in grain shape may involve an uncharacterized mechanism. In addition, the untargeted metabonomics analysis showed that only three overlapped differential metabolites, including key metabolites in amino acid metabolism L-Glutamine and 4-Hydroxycinnamic acid, were detected between ZH11-OE-1 *vs* ZH11 and ZH11-Cas9-1 *vs* ZH11 comparisons (Supplementary Fig. 10). The cell cycle genes play an important role in the process of cell proliferation, and expansins are plant cell wall proteins that are required for cell growth and elongation[17,18]. Amino acid metabolism is at the foundation of cell proliferation and growth[19]. Thus, we speculate that *GSE9* regulates grain shape probably by fine-tuning both the cell wall and cell cycle pathways, possibly

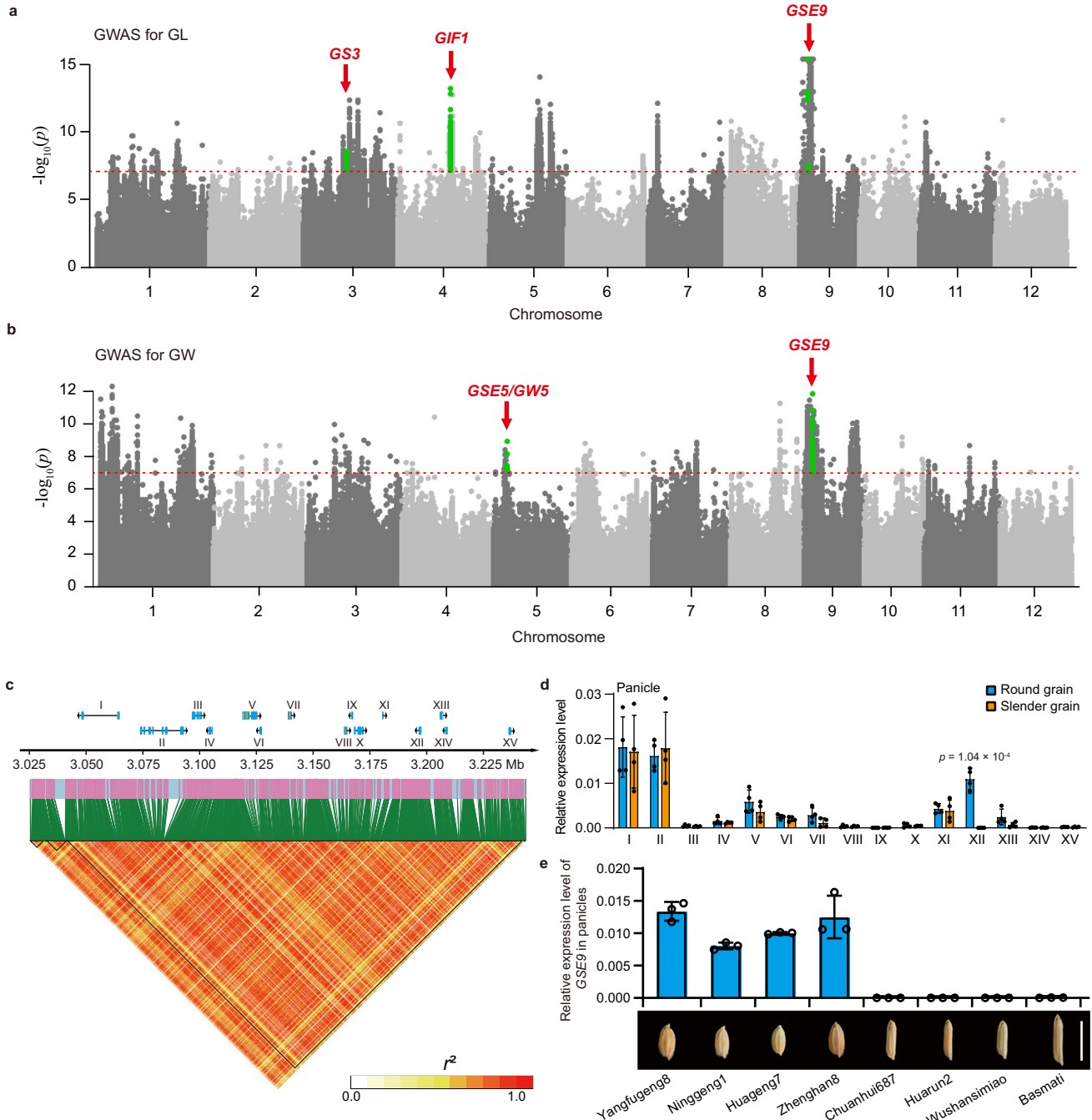

**Fig. 1 | Identification of the *GSE9* locus for grain shape by GWAS. a, b** Genome-wide association study of grain shape. Manhattan plots for grain length (GL) (**a**) and grain width (GW) (**b**). Dashed line represents the significance threshold ($P = 1 \times 10^{-7}$), which is determined by the Bonferroni correction method. Well-known loci for grain shape including *GS3*, *GIF1* and *GSE5/GW5* are indicated by red arrows. **c** LD heatmap of the region of *GSE9* locus and the genomic location of 15 predicted genes. Pairwise LD was determined by calculation of $r^2$ (the square of the correlation coefficient between SNPs). The 15 candidate genes are indicated by I to XV. **d** The expression levels of 15 candidate genes in the panicle of round-grain varieties and slender-grain varieties. *OsActin* was used as a control. Data show means ± SD ($n = 4$ biological replicates). **e** Expression analysis of the candidate gene XII (*GSE9*) in panicles from the selected varieties. Data show means ± SD ($n = 3$ biological replicates). Scale bar, 1 cm. Source data are provided as a Source Data file.

through an uncharacterized mechanism related to certain amino acid metabolic changes during cell proliferation and expansion.

## The *GSE9* gene originated from a previous non-coding region

To investigate the origin and distribution of *GSE9*, we performed comprehensive BLAST searches against numerous sequence databases. The results revealed that *GSE9* could not be detected in any taxa outside the genus *Oryza*, suggesting it is an orphan gene in *Oryza* (Supplementary Fig. 11). We further compared its syntenic regions

across the genus *Oryza*, and highly similar sequences were detected in *O. sativa*, *O. rufipogon*, *O. barthii*, *O. glumaepatula*, and *O. longistaminata*, but absent in *O. meridionalis*, *O. punctata*, and *O. brachyantha* (Supplementary Fig. 11c). The full coding sequence of *GSE9* was identified in *japonica* and Or-III type of *O. rufipogon*, but the other *Oryza* species including all AA-genome wild rice relatives do not contain the start codon site of *GSE9*, suggesting that the type without the start codon of *GSE9* was the ancestral state (Fig. 4a). The flanking genes of *GSE9* locus are conserved in physical location and collinear order in

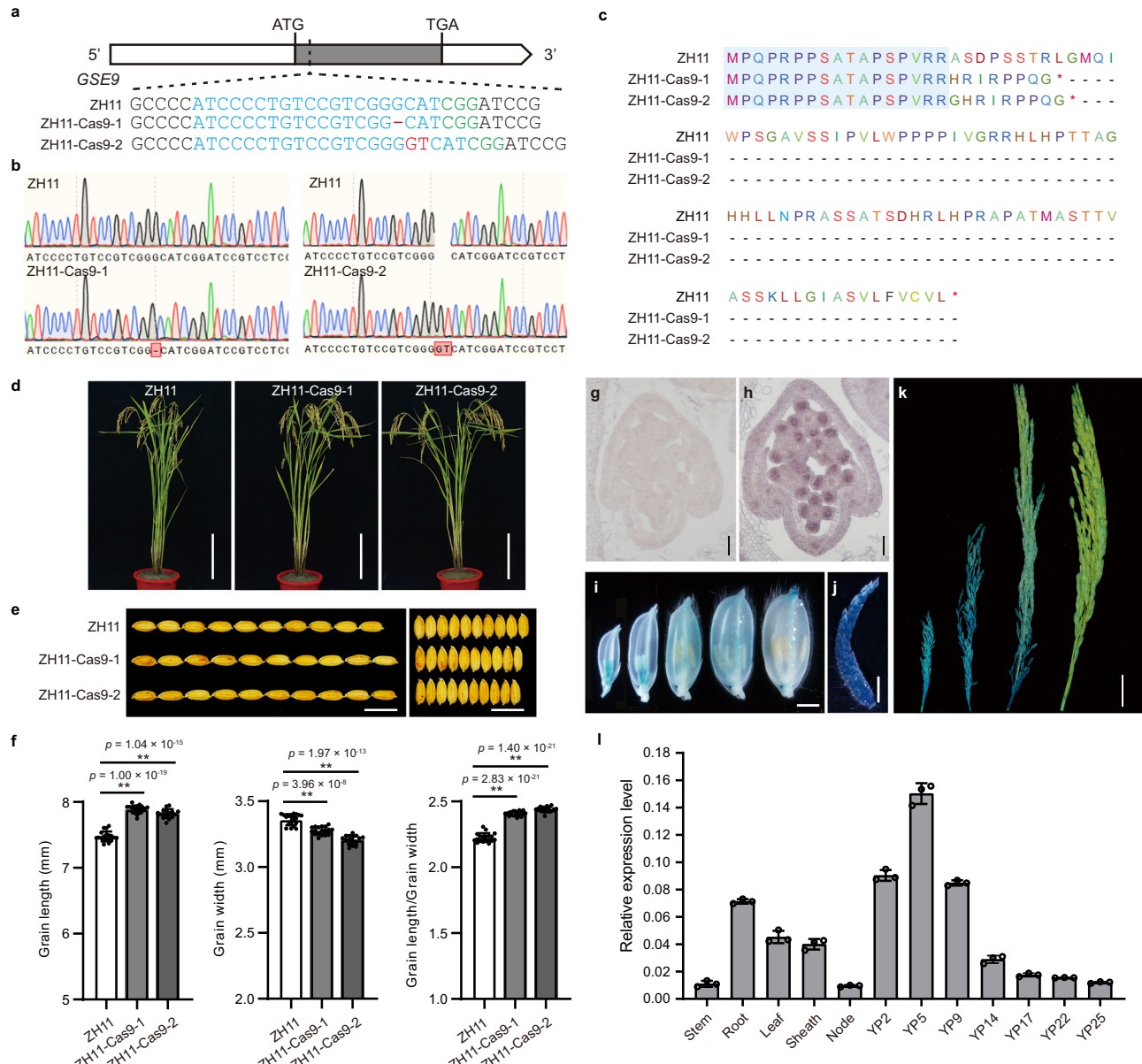

**Fig. 2 | Functional validation of the candidate gene *GSE9* in regulating grain shape. a–h** Identification of *GSE9* knockout mutants generated by the CRISPR/Cas9 system in Zhonghua11 (ZH11) background. **a** Targeted mutagenesis of *GSE9*. The target mutated sites are indicated on the gene structure of *GSE9*. Gray box indicates the single exon of *GSE9* gene. **b** Mutation events were confirmed by sequencing. The target mutated sites are marked with red box. **c** Comparisons of amino acid sequence encoded by *GSE9* gene in the wild-type ZH11 and the truncated sequence of GSE9 protein in *GSE9* knockout mutants. Red asterisk indicates the termination codon. **d–f** Phenotypic identification of *GSE9* knockout mutants in ZH11 background. **d** Plant morphology of ZH11 and *GSE9* knockout mutants at the mature stage. Scale bar, 20 cm. **e** Comparisons of grain shape between ZH11 and *GSE9* knockout mutants. Scale bar, 1 cm. **f** Statistical analysis of grain length, grain width, and grain length/width ratio between ZH11 and *GSE9* knockout mutants. Data show means ± SD (*n* = 19/18/16, 19/18/16, 19/18/16 biological replicates).

Statistical analysis was performed by two-tailed Student's *t*-test (**\**p* < 0.01). **g, h** In situ expression analysis of *GSE9* gene in spikelet hulls of the *japonica* variety. **g** Negative-control hybridization with *GSE9* sense probe. **h** In situ hybridization of *GSE9* in spikelet hulls. Scale bars, 50 μm. **i–k** *GSE9* expression activity was monitored using *GSE9pro::GUS* transgenic plants. Histochemical GUS staining in spikelet hulls (**i**), panicle of length 2 cm (**j**), and panicles of length 7, 14 and 22 cm (**k**). Scale bar, 2 mm in (**i–k**), 2 cm in (**k**). At least three independent replicates were performed and a representative result is shown. **l** qRT-PCR analysis of *GSE9* expression in various rice tissues. Stem, root, leaf, sheath, and node were harvested from ZH11 at the mature stage. Young panicles (YP) of different lengths (indicated as numbers, cm) were included for the analysis. *OsActin* was used as a control. Data shown are means ± SD of three biological replicates. Source data are provided as a Source Data file.

the genomes of *japonica* and *indica* (Fig. 4b). These findings indicated that *GSE9* was likely a de novo evolved gene from a previous non-coding region of common wild rice (*O. rufipogon*), through acquisition of the start codon. We further analyzed the selective constraints on the evolution of the open reading frame of *GSE9* genes. The high estimated $Ka/Ks$ ratio indicated a relatively high level of nonsynonymous substitutions in the *GSE9* (with ATG) and *gse9* (without ATG) reading frame

(Supplementary Table 4), implying that positive selection might have acted on the origin of this de novo gene.

Consistent with the sequence analyses, the transcript of *GSE9* could be detected in *japonica* and *O. rufipogon*, but not in *indica* and the other wild species (Fig. 4c, d and Supplementary Fig. 12). Transient transcription activity assays showed that the variations in the promoter region *GSE9* gene between *japonica* and *indica* did not result in

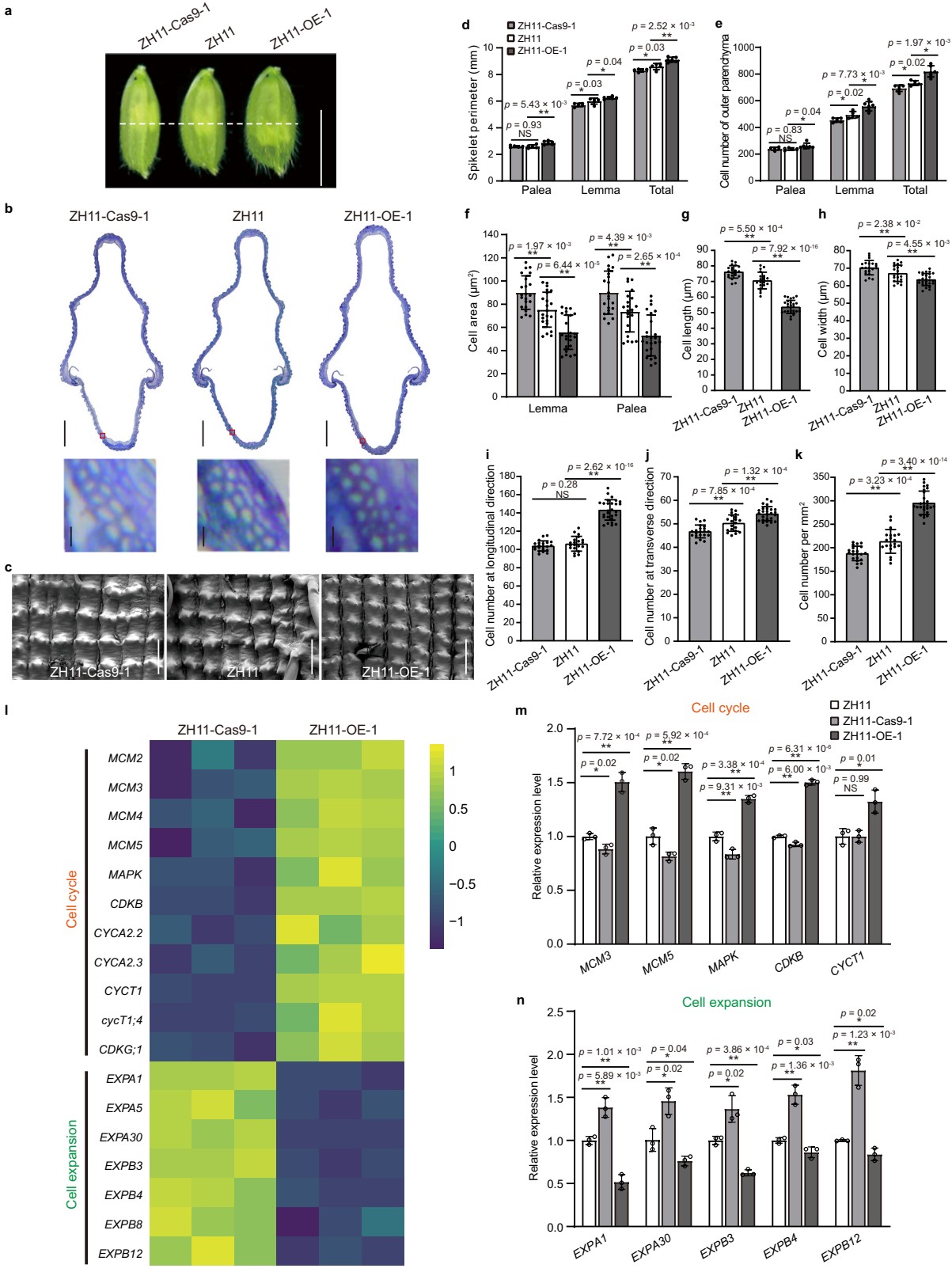

any significant difference of transcriptional activity, but higher transcriptional activity was detected in the sequence of *proGSE9 +* A than that of *proGSE9 +* G and *proGSE9* (Supplementary Fig. 13). Additional data revealed that different DNA methylation levels may result in the gene expression difference of *GSE9* between *indica* and *japonica*, while the G to A variation at the start codon of *GSE9* also contributes to DNA methylation difference in *indica* and *japonica* (Supplementary

Figs. 14–16). Thus, the most plausible explanation for these findings is that the G to A variation at the start codon of *GSE9* activated its transcription.

In addition to transcription, the protein-coding potential is a key factor for the function of de novo genes. A specific peptide with 27 amino acids of GSE9 protein was identified in *japonica* from proteomic database (ProteomeXchange ID: PXD001046) (Fig. 4a), suggesting that

**Fig. 3 | GSE9 controls grain shape by regulating cell number and cell size of the spikelet hull. a** Morphology of spikelet hulls of WT and *GSE9* transgenic lines before anthesis. The white dashed line indicates the localization of the cross-sections in (**b**). Scale bar, 5 mm. **b** Cross-sections of spikelet hulls for ZH11 and *GSE9* transgenic plants. Scale bar, 200 μm. The below-hand images show magnified views of the red boxed region. Scale bar, 20 μm. **c** Scanning electron microscopy analysis of spikelet hull outer surfaces of ZH11 and *GSE9* transgenic lines. Micrograph images provided were observed from at least three biological replicates and a representative result is shown. Scale bar, 100 μm. **d**–**f** Comparisons of spikelet perimeter (**d**), cell number (**e**), and cell area (**f**) of lemma and palea between ZH11 and *GSE9* transgenic plants. **g**–**k** Comparisons of cell length (**g**), cell width (**h**), cell number at the longitudinal direction (**i**), transverse direction (**j**) and per square

millimeter (**k**) of the spikelet hull outer surface of ZH11 and *GSE9* transgenic lines. The ZH11-Cas9-1 mutant showed larger cell size and fewer cells than ZH11, whereas the ZH11-OE-1 displayed the opposite phenotypes. **l**–**n** Differentially expressed genes related to cell cycle and cell expansion in ZH11-Cas9-1 mutant and ZH11-OE-1 line for *GSE9* in rice. **l** Transcriptional profile from RNA-seq data for a subset of genes related to cell cycle and cell expansion in ZH11-Cas9-1 mutant and ZH11-OE-1 plant. **m**, **n** Transcript levels of the selected genes were confirmed through qRT-PCR analysis with three biological replicates. *OsActin* was used as a control. Data show means ± SD (*n* = 5/5/6, 5/5/6, 5/5/6 cells in **d** and **e**; *n* = 22/22/24, 22/22/24 cells in **f**; *n* = 21/20/27 cells in **g** and **i**; *n* = 21/20/26 cells in **h**, **j** and **k**). Statistical analysis was performed by two-tailed Student's *t*-test (**p* < 0.01; **p* < 0.05; NS, not significant). Source data are provided as a Source Data file.

this gene is expressed at the protein level. Protein structure predictions showed only a single alpha helix in the C-terminus of GSE9 protein (Fig. 4e, f). Further 3D-BLAST of the Protein Data Bank (PDB), using the predicted GSE9 protein structure as a query, showed no significant similarity with proteins from any eukaryotic species (*E*-value cutoff = 1*e*-10). GSE9 protein also contained large intrinsic disordered regions (Fig. 4f). Western blot assays further confirmed that *GSE9* is a protein-coding gene (Fig. 4g). Given that *GSE9* encodes a protein without known sequence features, we also investigated the subcellular localization of GSE9 protein. GFP fluorescence observations revealed that GSE9 protein is predominantly localized to the plasma membrane (Fig. 4h, i), supporting the belief that new genes tend to encode putative membrane domains[20,21]. The above evidence clearly demonstrated that the de novo evolved *GSE9* is a protein-coding functional gene.

## Natural variations in *GSE9* contribute to the divergence of grain shape between *indica* and *japonica* subspecies

To further explore the evolutionary pattern of *GSE9* during rice domestication, we analyzed the genomic sequences of this gene in cultivated rice from 3 K rice genomes project[8], the Rice Super Pangenome Information Resource Database (RiceSuperPIRdb)[22] and our re-sequencing data[23]. We noticed that the distribution of A/G variations at the start codon site of *GSE9* was unbalanced between two rice subspecies (Supplementary Data 1 and 3). The vast majority of *indica* varieties do not possess the start codon at the *GSE9* locus due to the single nucleotide variation of A to G, whereas most of *japonica* varieties have the start codon site ATG (Supplementary Fig. 17). Thus, we defined the rice varieties carrying A and G variations at the start codon site of this gene as *GSE9* and *gse9* types, respectively. A geographic distribution analysis of 1,697 cultivated varieties from 3 K rice genomes project showed that the proportion of *GSE9* type has a trend of increasing along higher latitudes, while the *gse9* type was mainly present in lower latitudes with higher temperatures, implying a regional differentiation in *GSE9* (Fig. 5a). The estimated five parameter values of genetic divergence between two rice subspecies are relatively higher in *GSE9* locus than in its flanking genomic regions (Fig. 5b and Supplementary Table 5). The average $F_{ST}$ values in each 100-kb window were also estimated at the whole-genome level between *indica* and *japonica*, and our results revealed a high divergence (top 10% of $F_{ST}$ values) in the region of *GSE9* locus (Supplementary Fig. 18). All these findings suggested the genetic differentiation of *GSE9* between *japonica* and *indica* subspecies. The relative ratio of nucleotide diversity in *japonica* to wild rice revealed a selective sweep of ~1.2 Mb surrounding *GSE9* locus (Fig. 5c), which exhibited significantly reduced nucleotide diversity in *japonica* compared with wild rice (Supplementary Table 6), suggesting a strong artificial selection in *GSE9* locus of *japonica*. The estimated Tajima's *D* values in *GSE9* locus was significantly negative in *japonica* (Fig. 5d), implying directional selection across this region. By contrast, no obvious selection was detected in *indica* population because the relative ratio of nucleotide diversity in *indica* to wild rice was

higher than that in *japonica* to wild rice in *GSE9* locus. These results suggested that artificial selection might have contributed to the domestication of *GSE9* in the *japonica* population, which further facilitated the divergence of *indica* and *japonica*. We found that *GSE9* gene also diverged in the common wild rice. The start codon site of *GSE9* was absent in Or-I type, but present in Or-III type, which is consistent with its feature in *indica* and *japonica*, respectively (Supplementary Data 3). Further phylogenetic analysis showed that the clades of *indica* carrying the *gse9* type and *japonica* carrying the *GSE9* type nested within those of Or-I and Or-III, respectively (Fig. 5e and Supplementary Fig. 19). A haplotype network was also constructed to characterize the evolutionary relationships using all 58 haplotypes, and the haplotypes mainly present in *japonica* and *indica* varieties have the closest relationships with those in Or-III type and Or-I type of common wild rice, respectively (Fig. 5f). Given that *indica* and *japonica* have been suggested to be descended from Or-I and Or-III, respectively[24], we speculate that *indica* carrying the *gse9* type and *japonica* carrying the *GSE9* type were derived from Or-I and Or-III, respectively.

We further investigated the role of natural variation in the start codon of *GSE9* on grain shape using our re-sequencing data of *indica* varieties. Among them, 29 and 102 *indica* varieties carried *GSE9* and *gse9* types, respectively (Supplementary Fig. 17c). *GSE9* gene can transcribe in the *indica* varieties carrying the *GSE9* type, but not in those carrying the *gse9* type (Supplementary Fig. 20a). The *indica* varieties with the *gse9* type exhibited longer and narrower grains than those with the *GSE9* type (Supplementary Fig. 20b, c). These findings indicated that natural variation in the start codon site of *GSE9* has large effects on the grain shape of rice, which might be further exploited for the improvement of grain shape. In addition, we noticed that *O. rufipogon* accessions with the *gse9* type had longer and narrower grains than those with the *GSE9* type (Supplementary Fig. 21), consistent with the typical grain shape in cultivated rice of the *gse9* and *GSE9* types, respectively. Given that knockout of *GSE9* gene in *japonica* varieties led to slender grains (Fig. 2a–f and Supplementary Fig. 5), we then introduced the genomic region of *japonica* (NIP) *GSE9* into the *indica* variety Kasalath to explore the grain shape changes in the complementary lines of the *indica* background. The transcript of *GSE9* could be detected in the complementary lines of Kasalath (Supplementary Fig. 20d). Phenotypic analyses showed that the *GSE9*-complementary lines had no significant differences in multiple agronomic traits, but more round grains when compared with Kasalath (Supplementary Fig. 20e–g and Supplementary Table 7). These results clearly indicated the role of *GSE9* in grain shape difference between the two rice subspecies.

## Discussion

Our work uncovered that the de novo evolved *GSE9* gene is involved in grain shape difference between the two most widely cultivated subspecies of rice *indica/xian* and *japonica/geng* (Fig. 6). As an important driving force in evolutionary novelties, de novo evolved genes arise from DNA sequence that has not, at least recently, been protein-

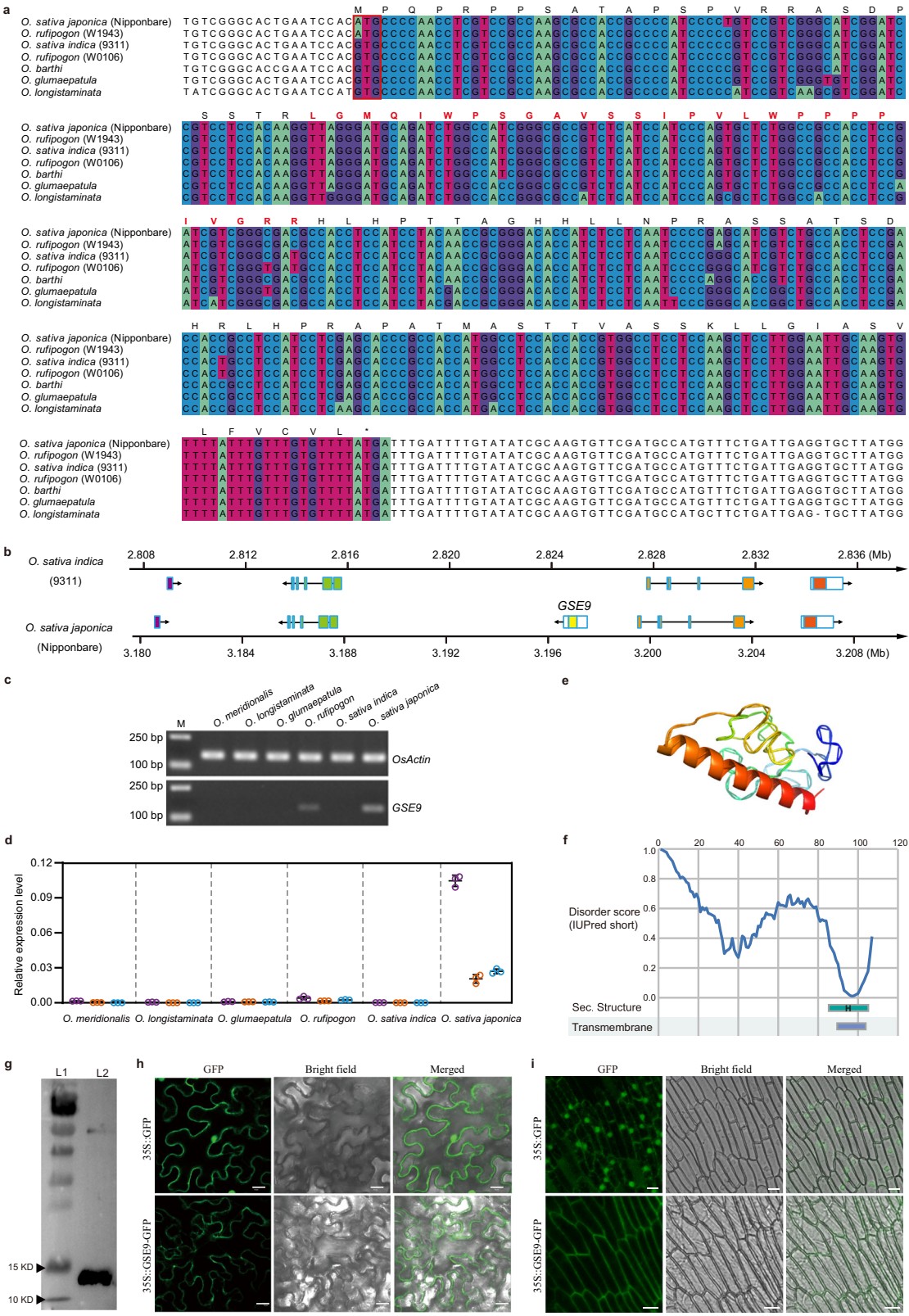

coding[25,26]. Thus, the most significant feature of de novo genes is that its syntenic regions in outgroup species lack the gene, i.e., the orthologous sequences of these genes exist, but do not encode an open reading frame[27]. Here, we found that *GSE9* is an *Oryza*-specific gene that is absent in related taxa outside the genus *Oryza*. The orthologous sequences of *GSE9* gene are present in the syntenic regions of several *Oryza* outgroup species, but they do not encode an ORF due to

absence of the start codon. By contrast, the full coding sequence feature of *GSE9* is present in *O. rufipogon* and most *japonica* varieties. In addition, our transient transcription activity assays, DNA methylation level analysis combined with the proteomic database analysis suggest that the G to A variation at the start codon of *GSE9* contributes to its de novo origination by acquiring the ability of both transcription and translation simultaneously. Thus, the most plausible explanation

**Fig. 4 | The *GSE9* is a protein-coding gene through de novo origination from a previous non-coding region. a** Sequence alignment of DNA sequence of *GSE9* gene and its homologous non-coding sequences in various *Oryza* populations. The nucleotides in the alignment of *GSE9* coding region are labeled by boxes of color. Red empty box indicates the start codon site of *GSE9* gene. Capital letters above the nucleotide sequences indicate the amino acids encoded by *GSE9* gene in *O. sativa japonica* (Nipponbare), and those labeled by red bold fonts indicate the specific peptides of GSE9 protein in ProteomeXchange (PXD001046). **b** Comparison of *GSE9* locus and its flanking sequences in *O. sativa japonica* and *indica*. **c** The transcript level of *GSE9* in the panicles of various *Oryza* species determined by RT-PCR. **d** qRT-PCR analyses of the *GSE9* transcripts in different tissues of selected *Oryza* species. Purple, orange and blue dots indicate the expression levels of *GSE9* in leaf, stem and panicle, respectively. *OsActin* was used as a control. Data shown

are means ± SD (*n* = 3 biological replicates). **e** Predicted 3D structures for the translation product of *GSE9* using QUARK. *GSE9* is predicted to encode a protein with an alpha-helix at the C-terminus. **f** Predicted sequence properties of GSE9 protein. Blue line plot showing the likelihood of disorder (IUPred score). Trans-membrane domains and predicted secondary structures (H: alpha-helix) are also shown. **g** Western blot analysis of His-GSE9 protein by anti-His antibody. L1: molecular protein standard; L2: purified His-GSE9 proteins. At least three independent replicates were performed and a representative result is shown. **h, i** Subcellular localization of *japonica* GSE9 protein in tobacco leaf cells (**h**) and root tissues of the transgenic plants (**i**). Micrograph images provided were observed from at least three biological replicates and a representative result is shown. Scale bar, 10 μm. Source data are provided as a Source Data file.

---

for these findings is that *GSE9* de novo evolved from a previous non-coding sequence of common wild rice, and was then inherited by *japonica*.

Most insights into de novo evolved genes come from large-scale comparative genome, transcriptome and proteome analyses thus far[2,5,28,29]. Although the putative functions of several de novo genes have been reported in certain eukaryotic species, including *Drosophila*[6,27] and *Arabidopsis*[30,31], the potential role of de novo genes remains unexplored in staple crops. None of these genes has been investigated in detail for their roles in morphological differentiation between subspecies. Our findings provided insights into the contribution of a rice de novo gene to the distinguishing traits (i.e., grain shape) between *indica* and *japonica* subspecies. The grain shape divergence may start in common wild rice with the *gse9* and *GSE9* types, and was then inherited by *indica* and *japonica*, respectively. Grain shape difference between *indica* and *japonica* is believed as a result of long-term natural and artificial selection[12,13,32]. In the present study, we noticed that *GSE9* locus conferring grain shape was under artificial selection in *japonica* rice, which may further contribute to its divergence between *japonica* and *indica*. Long grain shape is highly prized in rice market because of its excellent appearance quality[33], thus genetic manipulation of *GSE9* may have great potential in rice breeding application, especially in grain shape and yield production improvement. These findings broadened our understanding of the contribution of de novo genes in the genetic and morphological divergence between subspecies. We provide a gene target for the improvement of grain shape in rice. Nevertheless, further studies are needed to explore the detailed mechanism underlying the role of *GSE9* in grain shape.

## Methods

### Genome-wide association study

We carried out GWAS for grain shape using the panel of both *indica* and *japonica* varieties, which were obtained from the germplasm band or breeders in central China. We rephenotyped the grain shape-related traits, such as grain length (GL) and grain width (GW) of 289 rice varieties, including 158 *japonica* varieties genotyped previously[23] and 131 *indica* resequencing varieties. Tremendous variation in GL and GW was observed in these 289 rice varieties. A total of 2,269,131 single-nucleotide polymorphisms (SNPs) with a minor allele frequency (MAF) ≥ 0.01 among the whole rice genome of this resequencing population were used for GWAS under the general linear model in tassel v5.2.4[34]. Admixture software[35] was used to estimate the population structure. The *P*-value threshold for significance was set as $1 \times 10^{-7}$ by the Bonferroni correction method[36]. The region containing at least three consecutive significant SNPs was considered as a single associated signal, and the SNP with minimum *P*-value in the associated signal was determined to be the leading SNP. The linkage disequilibrium (LD, calculated as $r^2$) between SNPs in the 289 varieties was evaluated using plink v1.9[37]. The LD heatmaps surrounding peaks in the GWAS were constructed using the LDBlockShow v1.40[38].

Candidate genes were predicted within a genomic region of 220 kb of the leading SNP for QTL, and genes annotated as retrotransposon and transposon were excluded from the analysis.

### RNA extraction and quantitative reverse transcriptase PCR (qRT-PCR)

The RNA extraction kit (Tiangen Biotech, Beijing, China) was used to isolate total RNA from rice samples. The RNA was then reverse transcribed into the cDNA using the HiScript Q RT SuperMix for qPCR (Vazyme Biotech, Nanjing, China). qRT-PCR was conducted with the ChamQ SYBR qPCR Master Mix (Vazyme Biotech, Nanjing, China). The internal control gene *OsActin* (Gene locus: LOC_Os10g36650) transcripts were used to normalize the gene relative expression level.

### Rapid amplification of cDNA ends (RACE)

The full-length cDNA of *GSE9* was amplified by 5'- and 3'-RACE using the HiScript-TS 5'/3' RACE Kit (Vazyme Biotech, Nanjing, China). 1 μg total RNA from the *japonica* variety Zhonghua 11 was converted into 5'- or 3'-RACE-Ready first-strand cDNA using a 5' CDS Primer or a 3' CDS Primer (included in the kit), respectively. The cDNA was amplified using a universal primer mix (UPM) mixed with 5' or 3' gene specific primers (5'GSP-1, 5'GSP-2 and 3'GSP). The PCR products were then used as the template for nested PCR with the new gene specific primers (5'NGSP-1, 5'NGSP-2 and 3'NGSP). The amplified cDNA was cloned into the linearized vector using 5 min TA/Blunt-Zero Cloning Kit (Vazyme Biotech, Nanjing, China) and clones was then Sanger sequenced with M13F and M13R.

### Plasmid construction and genetic transformation

To generate the knockout mutants, two *GSE9* site specific single-guide RNAs were selected and introduced into the CRISPR/Cas9 system, in which the *Cas9* gene was under the control of *ubiquitin* gene promoter, while the expression of sgRNA was driven by the *U3b* promoter. To overexpress *GSE9*, its coding sequence was amplified and cloned into the construct pCAMBIAI1390 under the control of *ubiquitin* gene promoter. The *GSE9*-overexpression and knockout constructs were transformed into two *japonica* rice varieties Zhonghua11 (ZH11) and Nipponbare (NIP), respectively. To generate the complementation lines of *GSE9* gene in *indica* background, the coding sequence and 2-kb promoter region was amplified and cloned into the construct pCAMBIAI1301. The *GSE9* complementation construct was transformed into *indica* variety Kasalath. For promoter-GUS assay, a 2-kb genomic upstream region of the *GSE9* was amplified and cloned into the pCAMBIA1300 vector. For subcellular localization analysis, the coding sequence of *GSE9* was fused into the pCAMBIAI1300 vector with *GFP*. The *GSE9pro::GUS* and *GSE9::GFP* constructs were transformed into ZH11, respectively. All rice transformations were mediated by *Agrobacterium tumefaciens* strain EHA105. Primers used for plasmid construction are provided in Supplementary Data 4.

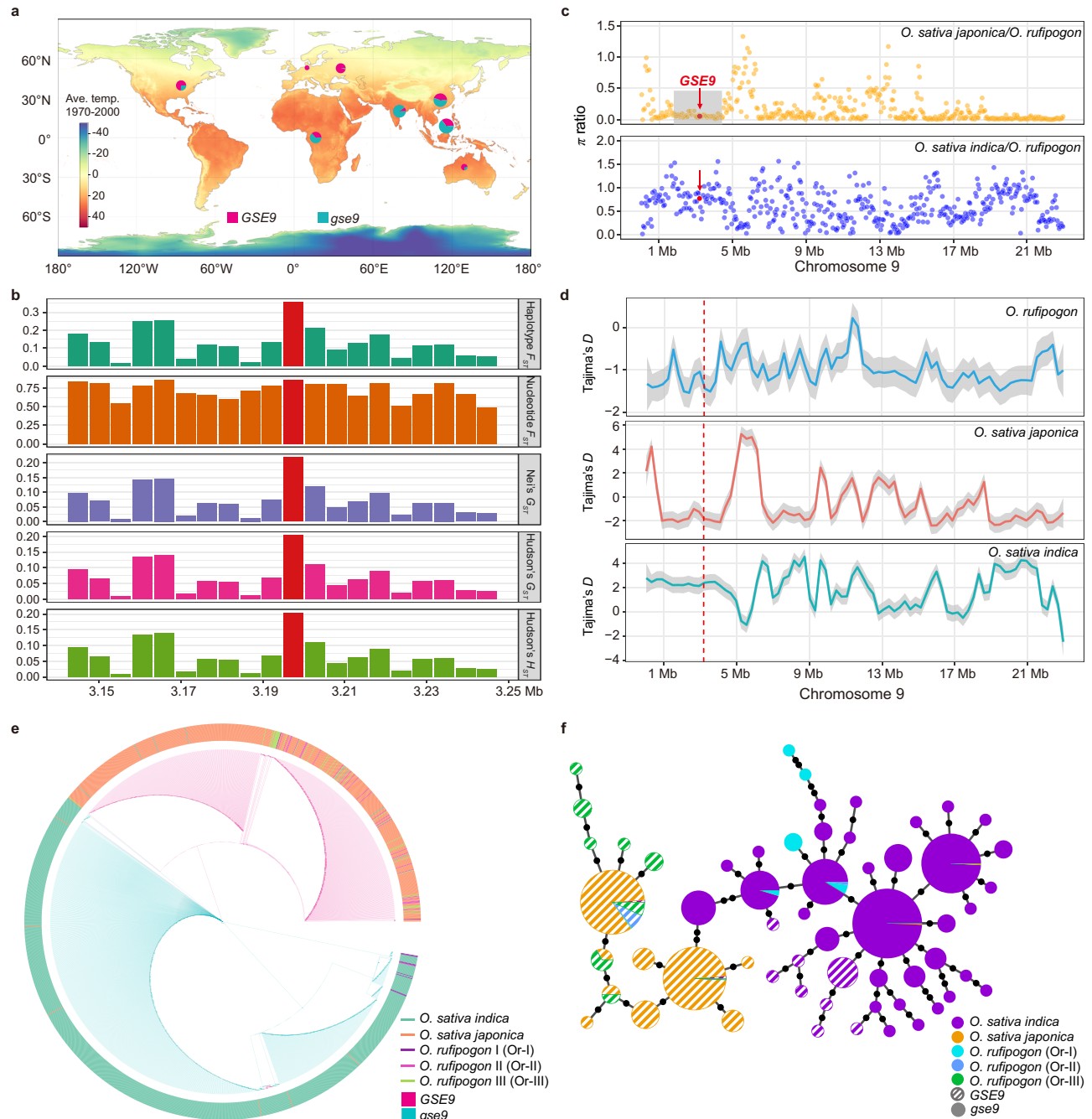

**Fig. 5 | Natural variation in *GSE9* contributes to the genetic divergence between *japonica* and *indica* subspecies. a** Geographic distributions of 1697 cultivated rice varieties. Red and blue circles indicate the *GSE9* and *gse9* type, respectively. **b** The estimated parameters of genetic differentiation between *japonica* and *indica* for *GSE9* locus and its flanking genomic regions. **c**, **d** The relative ratio of nucleotide diversity (**c**) and Tajima's *D* (**d**) analyses in the whole chromosome 9 of cultivated and common wild rice. Red arrows and dots indicate the *GSE9* locus. Gray shading in **c** indicates a selective sweep of ~1.2 Mb (2.95-4.15 Mb) region surrounding *GSE9* locus in *japonica*. Line in (**d**) represents the linear fitting, while gray shading indicates the 95% confidence interval. **e**, **f** Phylogeny (**e**) and haplotype networks (**f**) generated from the full-length cDNA sequence of *GSE9* in both cultivated rice and various groups of common wild rice varieties. Outer circle of the tree indicates various rice populations. Circle size of the network is proportional to the number of samples for each haplotype. Black spots on the lines indicate mutational steps between two haplotypes. Source data are provided as a Source Data file.

## GUS staining and GFP fluorescence observations
To determine the expression pattern of *GSE9* in detail, we sampled various rice tissues, including the young root, stem, sheath, leaf, node and young panicle from *GSE9pro::GUS* transgenic plants, and stained using a GUS staining kit (Coolaber Biotech, Beijing, China). The stained tissues were cleared with different concentrations of ethanol, and then observed under a microscope (OLYMPUS, MVX10, Japan).

The localization pattern of GSE9 was examined by imaging the root of 1-week-old seedlings of *GSE9::GFP* transgenic plants and *Nicotiana benthamiana* leaf epidermal cells using a LSM 710 confocal laser scanning microscope (Zeiss, Oberkochen, Germany).

## RNA in situ hybridization
Fresh young panicles were fixed in formaldehyde-acetic acid-ethanol solution, dehydrated in an ascending series of ethanol, and then

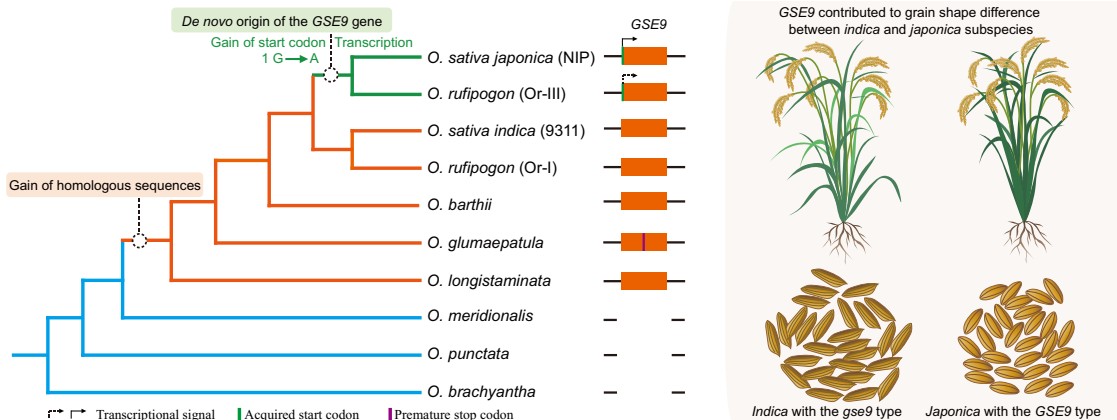

**Fig. 6 | De novo origination of *GSE9* contributes grain shape difference between *indica* and *japonica* subspecies.** The phylogenic tree indicates the origination process for the de novo evolved gene *GSE9*. The orthologous non-coding sequence of *GSE9* was present in *O. longistaminata*, *O. glumaepatula*, *O. rufipogon* I, and *O. sativa indica* (9311). A single nucleotide G to A substitution at the start codon site in the orthologous non-coding sequence of *GSE9* created the de novo ORF of this gene in *O. rufipogon* III that was then inherited by *O. sativa japonica* (NIP). The orange box indicates orthologous sequences of *GSE9* gene. The solid arrow indicates the strong transcriptional signal of *GSE9* gene in *japonica*. The dotted arrow

indicates the relatively low transcriptional level of this gene in *O. rufipogon* Or-III. The green bar shows the start codon acquisition of *GSE9*, while the purple bar indicates the premature stop codon in ORF of *GSE9*. Typical *indica* varieties exhibit slender grains, while typical *japonica* varieties show ovate grains. Knockout of *GSE9* led to slender grains in *japonica*, whereas introgression of this gene resulted in round grains in *indica* background. In the *indica* natural populations, the selected varieties with the *gse9* type exhibited slender grains than those with the *GSE9* type. All these observations revealed the role of *GSE9* gene in grain shape difference between *indica* and *japonica* subspecies.

embedded in paraffin wax. The blocks were sectioned into 8 μm slices by rotary microtome (Leica RM 2235, Germany) and mounted onto Poly-Prep slides (Sigma-Aldrich, St. Louis, MO, USA). A 388-bp specific fragment of *GSE9* cDNA was cloned into the pEASY-Blunt simple vector (Transgen Biotechnology). RNA probes were transcribed with T7 RNA polymerase and labeled by digoxigenin (Roche Diagnostics, Mannheim, Germany). The sections were hybridized to either a sense or an antisense probe of *GSE9*, where the sense probe was used as a negative control.

### Histological and cellular analysis

For the histological analysis, fresh young spikelet hulls before anthesis were fixed with 2.5% glutaraldehyde, dehydrated in a graded ethanol series, and then embedded in Technovit 7100 resin (Heraeus Kulzer, Wehrheim, Germany). The spikelet hulls were cut into 10-μm-thick sections by ultramicrotome (Leica EM, Germany), and observed using an electron microscope (Leica DM1000, Germany). The outer surfaces of the spikelet hulls were also observed using a scanning electron microscope (GeminiSEM 300, Carl Zeiss, Germany). ImageJ software was used to measure the cell size and cell area of epidermal cells.

### Transcriptome analysis

For transcriptome analysis, young panicles from ZH11 and *GSE9* transgenic plants before anthesis were collected and used for library construction and sequencing using the Illumina NovaSeq 6000 system. The generated reads were firstly filtered and then mapped onto the reference genome of *Oryza sativa* using HISAT2 v1.2[39]. The featureCounts software v2.0.1[40] was used to count the read numbers mapped to each gene. Finally, differentially expressed genes of knockout and overexpression lines compared to WT plants were identified using DESeq2 v1.36.0[41].

### Metabolite profiling analysis

Metabolites were extracted from young panicles with six biological replicates from ZH11 and transgenic plants. The untargeted metabolite profiling was performed using liquid chromatography-mass spectrometry (LC-MS) platform. Peak identification and metabolite identification were performed using the XCMS software v3.12.0[42]. The Variable Importance for the Projection (VIP) was calculated by the

orthogonal partial least squares discriminant analysis (OPLS-DA). VIP > 1 and *P*-value < 0.05 were used as the criteria to determine the differential metabolites among ZH11-OE-1 *vs* ZH11 and ZH11-Cas9-1 *vs* ZH11 comparisons.

### Purification and expression of GSE9 protein

The coding sequence of *GSE9* was inserted into the pCZN1 vector carrying 6 x His tag protein. The recombinant pCZN1-GSE9 fusion protein was then expressed in *Escherichia coli* strain BL21. Protein expression was induced by adding 0.4 mM isopropyl-β-thiogalactopyranoside (IPTG) for 4 h at 37 °C. For the western blot assay, the purified recombinant fusion protein was separated using SDS-PAGE, and subsequently transferred to PVDF membranes to detect His protein after electrophoresis. Detection of GSE9 proteins was performed using anti-6x His antibody (Sigma-Aldrich, SAB2702220, 1:8000 dilution) and goat anti-mouse IgG-HPR secondary antibody (CWBIO, CW0102S, 1:10000 dilution).

### Sequence homology detection and protein structural prediction

To detect homologs of *GSE9* (Phytozome identifier LOC_Os09g06719), its genomic or protein sequence was used as query to perform BLAST searches against NCBI *nr* and *est* database, Phytozome, the One Thousand Plant Transcriptome (OneKP) database[43], Gramene[44], OryzaGenome[45], Rice Functional Genomics and Breeding Database (RFGB)[46], RiceSuperPIRdb[22] and other relevant databases (*E*-value cutoff = 1*e*-5). Additional pHMMER searches were performed against References Proteomes (*E*-value cutoff = 1*e*-5).

The ab initio tertiary structure prediction of GSE9 protein was performed using QUARK server[47]. The top predicted model from QUARK was displayed by PyMOL. PSIPRED web server (http://bioinf.cs.ucl.ac.uk/psipred/) was used for the prediction of secondary structure. The IUPred short algorithm[48,49] was used to predict intrinsic disorder of GSE9 protein sequence.

### Transient dual luciferase (dual-LUC) assay

To generate the NIP-*proGSE9*::LUC and MH63-*proGSE9*::LUC constructs, we amplified approximately 2-kb promoter region upstream of the *GSE9* locus from the *japonica* variety Nipponbare (NIP) and the *indica* variety Minghui 63 (MH63), and then ligated them to the

pGreen0800-LUC vector. We also constructed NIP-*proGSE9* + A::LUC, NIP-*proGSE9* + G::LUC, MH63-*proGSE9* + A::LUC and MH63-*proGSE9* + G::LUC vectors, respectively. Transient transactivation assays were performed using the tobacco (*Nicotiana benthamiana*) leaf system. Firefly luciferase activities derived from the various constructs and *Renilla* luciferase under the control of the 35 S promoter were quantified using the Dual-Luciferase Reporter Assay kit (Vazyme Biotech, Nanjing, China).

## DNA methylation level analyses

To detect the DNA methylation level of the promoter region of *GSE9* locus, we isolated DNA using cetyltrimethylammonium bromide (CTAB) from several cultivated rice varieties, including the *japonica* varieties Nipponbare (NIP), Zhonghua11 (ZH11), Wuyunjing7 (WYJ7) and Dongjing (DJ), and the *indica* varieties 9311, Minghui63 (MH63), Kasalath, Huanghuazhan (HHZ) and Guangluai4 (GLA4). The isolated DNA was digested by the methylation sensitive (HpaII) and insensitive (MspI) restriction enzyme pairs. The digested DNA was then used as a template to amplify the promoter fragment of *GSE9* locus that contains three CCGG sites. We also performed searches against rice methylation database RiceENCODE[50] to detect the DNA methylation levels in the genomic region of *GSE9* in the *indica* variety (MH63) and *japonica* variety (NIP).

## Population genetic and evolutionary analyses

Genomic sequences of 1,697 cultivated and 76 wild varieties were obtained from RFGB and OryzaGenome, respectively. The distribution of two types of *GSE9* with or without start codon was compared between *japonica* and *indica* varieties using chi-square ($\chi^2$) test. The geographical information of cultivated groups was obtained from RFGB, and marked on map using *Cartopy* package v0.20.0[51] in Python v3.6.0 software to observe geographic distribution of the two types. The map layer of terrestrial temperature averages from 1970 to 2000 A.D. was obtained from WorldClim[52]. The genetic difference parameters were estimated using *PopGenome* package v2.1.6[53] in R v4.1.2 software within 5-kb sliding windows between *japonica* and *indica* groups. The average $F_{ST}$ values in each 100-kb window were also estimated at the whole-genome level between *indica* and *japonica* subspecies using VCFtools v0.1.16[54]. The nucleotide diversity ($\pi$) and Neutral test (Tajima's $D$) of each population were calculated in 50-kb windows using VCFtools v0.1.16[54]. Phylogenetic trees and haplotypes networks were inferred for all sites with minor allele frequency ≥0.01 in this locus. The phylogenetic tree for the full-length cDNA sequences of *GSE9* gene and their homologs was constructed using a maximum likelihood (ML) method by IQ-TREE v.2.1.2[55]. In the IQ-TREE, ModelFinder[56] was used to determine the optimal model of amino acid substitution and rate heterogeneity. According to the BIC score, of 546 protein models, K3Pu+F + I was chosen to construct a ML phylogenetic tree, and the branch support values were calculated by ultrafast bootstrap[57] and Shimodaira–Hasegawa-like approximate likelihood ratio test (SH-aLRT)[58] using 1,000 replicates. The haplotype network was calculated using *pegas* package v1.2[59] in R v4.1.2 software, and then displayed using plotting module matplotlib v3.6.0[60] by Python v3.6.0 software.

## Data analysis

Data are presented as the means ± standard deviation (SD), shown by error bars. The chi-square ($\chi^2$) test and independent-samples *t*-test (\**p* < 0.05; \*\**p* < 0.01; NS, not significant) were used for statistical analysis using IBM SPSS software.

## Primers

All primers used in this study are provided in Supplementary Data 4.

## Reporting summary

Further information on research design is available in the Nature Portfolio Reporting Summary linked to this article.

## Data availability

Gene resequencing data are available in the European Variation Archive (EVA) at EMBL-EBI under accession number PRJEB65579. RNA-seq data generated as part of this study have been deposited to the NCBI GEO database under the BioProject accession number GSE218565. The full-length cDNA of *GSE9* sequence confirmed by RACE is available through NCBI GenBank under accession number OR050540.1. The genomic sequences of *GSE9* in 1697 cultivated rice varieties were downloaded from the Rice Functional Genomics and Breeding Database (RFGB) (http://www.rmbreeding.cn/). The genomic sequences of *GSE9* in wild rice accessions were downloaded from OryzaGenome (http://viewer.shigen.info/oryzagenome2detail/index.xhtml) or Gramene (https://www.gramene.org/). The A/G variations in *GSE9* start codon site of 192 *O. sativa*, 19 *O. rufipogon* and 8 *O. barthii* accessions were obtained from the Rice Super Pan-genome Information Resource Database (RiceSuperPIRdb) (http://www.ricesuperpir.com/). The DNA methylation levels in the genomic region of *GSE9* were obtained from RiceENCODE (http://glab.hzau.edu.cn/RiceENCODE/). The genetic materials used in this study are available from the corresponding authors upon request. Source data are provided with this paper.

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

## Acknowledgements

This work was supported by grants from the National Natural Science Foundation of China (32070558, Recipient: Z.Y.; 32061143030, Recipient: C.X.; 32100448, Recipient: Y.L.; 32170636, Recipient: Y.X.; 31972487, Recipient: P.L.;), Natural Science Foundation of Jiangsu Province (BK20210799, Recipient: Y.L.), A project Funded by the Priority Academic Program Development of Jiangsu Higher Education

Institutions (PAPD), the Seed Industry Revitalization Project of Jiangsu Province (JBGS[2021]009, Recipient: C.X.), the Key Research and Development Program of Jiangsu Province (BE20223439, Recipient: Y.X.), and the Shanghai Science and Technology Agriculture Project ([2022] No. 1-6, Recipient: Z.Y.), and the Project of Zhongshan Biological Breeding Laboratory (BM2022008-029, Recipient: Z.Y.).

## Author contributions

Z.Y., A.Li., J.H. and C.X. conceived the idea and supervised this study. R.C., N.X., Y.L., T.T., Q.H., S.W., Z.W., M.C., Q.B., Z.L., H.W., Y.S., Y.J., J.D., A.G., and H.L. performed the experiments. R.C., N.X., Y.L., Y.Z., S.T., G.L., H.Z., C.Y., X.Z., Z.C., Y.X., and P.L. participated in the data analysis. R.C., N.X., Y.L., and Z.Y. wrote the original draft of the manuscript. All authors reviewed and edited the final draft.

## Competing interests

The authors declare no competing interests.

## Additional information

**Peer review information** : *Nature Communications* thanks Peter Civáň, Kousuke Hanada and the other, anonymous, reviewer(s) for their contribution to the peer review of this work. A peer review file is available.

