## [Peer Review File · Nature Communications]

A de novo evolved gene contributes to rice grain shape difference between indica and japonicaReviewers' Comments:

Reviewer #1:

Remarks to the Author:

The manuscript "A naturally evolved de novo gene contributes to rice grain shape difference between indica and japonica" identified a gene GSE9 controlling grain shape difference between indica and japonica. Most japonica varieties contain the GSE9 gene. However, the majority of indica varieties absence the start codon. Population evolutionary analyses revealed that gse9 and GSE9 were derived from Or-I and Or-III type of wild rice, respectively. GSE9 de novo evolved from a previously noncoding region of wild rice *Oryza rufipogon* through the acquisition of start codon. The findings provided new molecular evidence for the de novo gene contributes to the genetic and morphological divergence between the two rice cultivated subspecies, and provide a new target for precise manipulation of rice grain shape. I have following comments :

- 1.The authors present several data supporting their finding that the key mutation between GSE9 and gse9 was a SNP which resulting in the acquiring of start codon. The GSE9 is only expressed in the japonica varieties. In my opinion, it is odd that why this SNP was responsible for the expression difference between GSE9 and gse9? Are there any sequence differences in the promoter region of GSE9 between GSE9 and gse9. Oddly, despite doing many transgenic rice lines they do not produce a line where only the single SNP is introduced in modern rice, that in accordance with the previous finding would support their hypothesis. I think that would strengthen significantly their paper and would provide a very nice example of how a small change in the DNA can evolve de novo gene contributes to rice grain shape difference between indica and japonica.
2. The authors didn't tell us what exactly GSE9 encodes and how it could affect the cell proliferation and expansion. It is necessary to be addressed for publishing in high impact Journal.
- 3.In FIG2, the authors found that knock outing GSE9 in japonica varieties can increase the seed length. However, it also decrees the seed width. How about the seed weight of the mutant and wild? It is important for the assessing the application potential of GSE9.
- 4.The detail gene structure and the DNA and protein sequences are not found in the manuscript. Since GSE9 is a new gene without full length cDNA, it is better to carry out RACE to confirm the full length of the gene.
- 5.Phenotypic analyses showed that the GSE9-complementary lines had no significant differences in multiple agronomic traits, but more round grains when compared with that of Kasalath. However, in the FIG6, It seem the tiller number is different in the indica and japonica.
- 6.The Fig3b are not clear and the legend is two simple, which not mentioned the difference between ZH11 and the mutant. In Fig3d it is better to using mm for the Spikelet perimeter.

Reviewer #2:

Remarks to the Author:

In my view, the manuscript 'A naturally evolved de novo gene contributes to rice grain shape difference between indica and japonica' combines two layers of information. The first one is the GWAS-based identification and experimental validation of the GSE9 gene that contributes to the rice grain morphology. The second layer relates to the claim that this is a 'de novo gene', and provides some population genetic statistics. The identification and validation of the GSE9 gene presents some nice experimentation, including transcript analysis, genetic transformation with CRISPR/Cas9, cellular localization of GSE::GFP in *Nicotiana*, and more. For this part, I only have a couple of minor points.

1. The GWAS analysis identified quite a few peaks beyond the chosen significance threshold. Some of these apparently correspond to genes known to be involved in grain size and shape, and are indicated on Fig. 1a,b. However, these genes are not mentioned in the Figure legend, and it would be appropriate to mention them in the introductory paragraph, too. Furthermore, besides these known genes and the GSE9 peak, there are several additional signals, and it is not quite clear why the authors focused on the GSE9 signal. Is this the only signal that is detected for both grain length and width? Even so, why are the authors focusing only on genes that affect length and width

simultaneously and disregard signals that affect length and width separately? An explanation of the thought process would be nice.

2. A related comment is about a nuanced phrasing of the role of GSE9. The abstract says that GSE9 'controls' grain shape difference between indica and japonica. However, the CRISPR/Cas9 experiments showed that the grain length and width changed only about 5% and 3.5%, respectively. Since the median/modal differences between indica and japonica are much larger (shown on Supplementary fig. 1), it should be clearly formulated that GSE9 is only one of the genes that contribute to this difference.

The second layer of the manuscript is a dip into some murky waters. The authors claim that GSE9 is a 'de novo gene', i.e. it originated from a non-coding sequence. One fundamental problem with this statement is that depending on how far back in the molecular evolution we go, ultimately all genes originate from non-coding sequences. So what is the definition of a 'de novo gene'? I thought that perhaps the authors are trying to say that the gene originated in cultivated rice, i.e. after domestication. However, this is still not clear to me, as the text is a bit ambiguous on that. Was the GSE9 haplotype with the start codon found in wild rice? Parts of the text imply that the start codon was acquired in *O. rufipogon* (lines 41-42, 137-138, 142-144), but the haplotype network does not show wild rice of the GSE9 type.

The authors have demonstrated that some rice genotypes have the start codon and express GSE9, while others lack the start codon and do not express GSE9. But how do we now what is the ancestral state? Perhaps the ancestral state is the ATG start codon, and the loss of it is the derived state, in which case we are looking at a pseudogenization rather than a de novo gene emergence. Absence of GSE9 homologs outside of the genus *Oryza* indicates that the gene is relatively young, but it does not tell us whether that particular ATG is derived or ancestral. The haplotype network (Fig. 5f) could be informative in this respect, but is under-interpreted by the authors. The GSE9 types (I assume it means with the ATG start codon) are scattered all over the network, without showing a single origin. I can think of three possible explanations. (i) the ATG variant emerged multiple times in different haplogroups through convergent evolution (the network suggests at least 6 times, which seems unlikely). (ii) the ATG variant is very old, and the network suffers from poor sampling of wild diversity, which could place the GSE9 type into all central nodes. (iii) the haplotype network includes recombined sequences, i.e. not true haplotypes. It is not clear to me what region was used for the construction of the network (and the dendrogram). Was it just the gene sequence, or a wider region? I think the genealogy of the haplotypes needs to be analysed more carefully. Additionally, I would like to see a Ka/Ks analysis of the GSE9 and *gse9* reading frames. If it turns out there is an excess of synonymous substitutions in the *gse9* reading frame, that would indicate that the reading frame used to be a functional gene before it lost its start codon.

I have some additional minor comments:

- lines 106-107: 'GSE9 was the most likely candidate gene for GSE9' should be corrected to 'LOC_Os09g06719 was the most likely candidate gene for GSE9', and similarly in the whole paragraph.

- lines 166-170: The two sentences are contradictory. The sequences cannot be simultaneously evolving neutrally and result from natural selection. The low Pi does not necessarily indicate selection (it can have several reasons).

- lines 2010-211: It should be rephrased. The text does not really provide a strategy.

Peter Civan, 24 Jan 2023

Reviewer #3:

Remarks to the Author:

In this manuscript, Chen and co-authors describe a new rice de novo gene, GSE9, which encode a functional protein in the variety *Oryza sativa japonica/geng*, whereas the orthologous sequence *gse9*

is not expressed in *Oryza sativa indica/xian* and other closely related *Oryza* species. Through a series of thorough experimental assays, the authors demonstrate that GSE9 is involved in determining the grain shape differences between the japonica (ovate grains) and indica (slender grains) varieties. Comparative genomic analyses performed here showed that GSE9 is characterized by an enabling mutation that established a new start codon. Most japonica varieties contain this start codon, as opposed to a very few indica varieties. Using population and comparative genetic data the authors suggest that GSE9 the type OR-III of the wild rice *Oryza rufipogon*.

Increasingly, de novo gene birth is described as one of the most fundamental and common processes that originate novel genes across eukaryotes. Although de novo genes are heralded as potentially major contributors to phenotypic variation and evolutionary innovations, the functional impact of de novo genes has been thoroughly assessed only in a limited number of cases. This work represents one of the most comprehensive analysis of the functional role of de novo genes ever conducted, and the most significant contribution to dissecting the origin and phenotypic impact of de novo genes in monocots, with a comparable importance to the only other well characterized de novo gene in plants, the *Arabidopsis thaliana*-specific gene QQS. The possible application of this discovery to rice grain shape manipulations is noteworthy.

The methodological approach is sound and the data presented are convincing, although not comprehensive, as detailed below.

My major concern about the paper is about the association between the emergence of the start codon that characterize GSE9 with the expression and functionality of GSE9 itself. It is my opinion that the body of evidence provided in the manuscript does not rule out the possibility that a functional GSE9 protein evolved before the formation of the canonical start codon that is common in *O. sativa japonica*.

According to the sequences shown in fig 4a, most of the open reading frame of GSE9 is also present in the orthologs from the other species except in *O. glumipatula* and *O. rufipogon* (W0106), which share a stop codon around the middle of the ORF (CGA to TGA).

O. rufipogon (W1943), *O. sativa indica* (9311), *O. barthi* and *O. longistaminata* orthologs to GSE9 encode for a putative protein with a start codon at position 85 of the GSE9 codon (MQIWP..), as a quick BLAST search confirmed. These putative proteins are very similar to the GSE9 protein and include the region that contains the transmembrane domain/alpha-helix shown in fig. 4d. This suggest that a certain degree of conservation of the GSE9 protein occurs across multiple *Oryza* species, which share the ability to encode a shorter, but possibly functional, protein.

Because *gse9* is not transcribed in the panicles of non-japonica *Oryza* species and varieties (fig. 4e), there is no indication whether proteins encoded by *gse9* genes are indeed functional.

This does not contradict the main conclusions of the manuscript. However, it suggests that the evolutionary history of GSE9 might be more complicated than proposed in the article. In particular, it is possible that the C to A change that established the GSE9 gene start codon was not instrumental to encode a functional protein. A potentially more impactful change seems to be the transition from a non-transcriptional (or very low transcriptional) state to a transcribed coding sequence that occurred in *O. sativa japonica*, and to a much smaller extent in *O. rufipogon*, according to fig. 4e.

Additional data do not provide support to the claim that the GSE9 new start codon is responsible for the acquisition of functionality by the GSE9 protein.

For instance, the Extended Data Fig. 3a shows that the *O. indica* varieties with the GSE9 gene type, but not the *gse9* type, are transcribed. The authors suggest that "These findings indicated that natural variation in the start codon site of GSE9 has large effects on the grain shape of rice, which can be further exploited for the improvement of grain shape." However, this can also be explained by regulatory changes linked to the start codon site of GSE9.

Indeed, the complementary lines of Kasalath were built using a construct that contains the putative promoter of GSE9. Lines 250-252: "To generate the complementation lines of GSE9 gene in indica

background, the coding sequence and 2-kb promoter region was amplified and cloned into the construct pCAMBIA11301."

It is therefore possible that the regulatory changes responsible for the expression of the GSE9 are more functionally important than the C to A mutation establishing the start codon to produce a functional protein by enabling the transcription of this gene.

The authors suggest that "indica carrying the gse9 type and japonica carrying the GSE9 type were derived from Or-I and Or-III, respectively.", based on figs. 5e and 5f phylogenetic and haplotype data. I find this interpretation debatable. There are too few *O. rufipogon* samples to very clearly address the question of where the GSE9 type evolved. Looking at Fig. 5e., I don't see "that the clades of indica carrying the gse9 type and japonica carrying the GSE9 type nested within those of Or-I and Or-III, respectively (Fig. 5e)." This could be due to the quality of the figure, but it appears to be primarily an issue with undersampling of *O. rufipogon*. Furthermore, the phylogeny show no statistical support values for potentially key nodes supporting the 'nesting' of clades. In fig. 5f, some Or-III haplotypes are more closely related to indica than japonica, but it's not stated if these indica haplotypes contain GSE9 or gse9.

Fig. 6. This figure could be inaccurate if experiments show that GSE9 expression and protein functionality do not depend on the establishment of the new start codon. Also, expression in *O. rufipogon* is much lower—barely present, really—than in japonica. Furthermore, it would be important to indicate if grain shape in Or-I and Or-III mirror the shape of japonica and indica, respectively.

Suggested improvements. Given the consideration above, the following tests/experiments are recommended. A construct containing the 2kb of the promoter region but without the start codon found in most japonica varieties should be tested in a japonica line, and phenotypic data collected to show if the new start codon is indeed required to produce the observed phenotype of the wild-type GSE9 gene.

Additional comparative genomic analyses should be presented to show if the promoter region of the GSE9 gene contains unique mutations compared to the gse9 orthologs, and if so, the mutated regions should be also tested via one or multiple constructs to verify their possible role in enabling the transcription of GSE9. Epigenetic changes could also be responsible for the expression of GSE9 and need to be discussed in the paper.

Data from additional *O. rufipogon* varieties should be included, and if not available this should be discussed in terms of the possible scenarios of GSE9 evolution.

Data on *O. rufipogon* grain shape for the varieties Or-I and Or-III should be added.

Minor comments.

Title. I am not sure what the authors mean by 'A naturally evolved'. It may be more appropriate to indicate the time of origin of this gene, i.e. 'Oryza sativa-specific'.

Line 56: 'forece' should be 'force'

Line 57. 'modification of existing genes' is not a process of new gene evolution. Modification implies changes of the current status, i.e. sequence, of a gene. New but non-denovo genes evolve via duplication, TE exaptation, chimerism, horizontal gene transfer and other processes that are not quite 'modifications', except maybe chimerism. This should be rephrased.

Reference 2 seems misplaced or not appropriate, there is no mention of de novo genes and only one reference about gene duplication, while it contains an extensive description of gene regulatory

network changes influencing evolution. However, these changes have nothing to do with the emergence of new genes. Reference 3 is a recent excellent review of de novo genes that should be cited in Line 57.

Lines 160-63. Given the low number of locations, the association between latitude and GSE9/gse9 is inconclusive. Also, analyses based on climate variables and niche reconstructions are better suited to investigate this aspect than crude latitudinal data.

Fig. 5. The colors of GSE9 and gse9 appear to be switched between 5a and 5e.

In Figure 5e I found it difficult to point to where Or-I and Or-III are in the tree, these specimens should be better highlighted by different colors or labels on the tree.

Reviewer #4:

Remarks to the Author:

Comments on "A naturally evolved de novo gene contributes to rice grain shape difference between indica and japonica" by Rujia Chen et al.

The authors inferred a de-novo gene associated with rice grain shape in japonica by GWAS analysis. The knock-out analyses candidate genes perfectly showed the expected results. Therefore, I think that this manuscript is deserved for publication in a great journal. It is my great honor to review this manuscript. However, I think that this manuscript contains several minor problems of evolutionary analyses. Therefore, I recommend that the authors improve evolutionary analyses.

The followings are my comments:

Major comments

1) This manuscript did not perform robust analyses in population genetics. In Fig. 5b, the authors did not see any statistical test to identify the difference japonica and indica subspecies. Also, the authors should infer such the values in genomic scale. The values inferred from genomic scale can be values assuming neutral evolution. Please compare the values of GSE9 flanking regions with those assuming neutral evolution.

2) In Fig. 5c, I cannot understand the meaning of "low nucleotide diversity (π) for the GSE9 locus in both cultivated and wild rice might be the result of natural selection". Again, the authors did not perform any statistical test. I think that the authors would like to identify selective sweep in japonica but no selective sweep in indica. If so, the author can infer difference of nucleotide diversity between japonica and indica in GSE9 locus. The difference of nucleotide diversity can be compared with the nucleotide diversity in neutral evolution. The nucleotide diversity of neutral evolution can be inferred from genomic scale analysis. I think that you can do the similar analyses in a Genome Research paper (Shirai et al., Positive selective sweeps of epigenetic mutations regulating specialized metabolites in plants. 2021). The authors can try to compare nucleotide diversity in population either with or without ATG in GSE9. I'm curious to see selective sweep in japonica.

3) In Fig. 5d, the authors described that "none of Tajima's D 169 values for all estimated populations were statistically significant (Fig. 5d), implying 170 that GSE9 locus did not escape from neutral evolution during rice domestication." I think that the authors should infer Tajima's D values in japonica with ATG, japonica without ATG, indica with ATG and indica without ATG. The authors then examine the differences of Tajima's D values among different populations.

Minor comments.

1) The outgroup species of japonica and indica is likely to be *Oryza rufipogon* (Or-III) and *Oryza rufipogon* (Or-I), respectively. However, it is hard to identify the relationship from Fig. 5e. Please

change the colors to distinguish indica, japonica, Or-I, Or-II and Or-III in Fig. 5e easily. Furthermore, the relationship was addressed in some of previous analyses (Huang et al., Nature 2012). Therefore, please cite such the previous papers to enhance the evolutionary relationship.

2) Although the authors concluded "Remarkably, none of the plant de novo genes have been shown the roles in phenotypic variation and morphological differentiation", several papers introduced de-novo genes in plants. Please cite these papers as follows.

1. Takeda et al. A de novo gene originating from the mitochondria controls floral transition in *Arabidopsis thaliana*. Takeda et al., *Plant Molecular Biology* 2022.

2. Mingsheng et al., QQS orphan gene and its interactor NF-YC4 reduce susceptibility to pathogens and pests. 2018.

Dear reviewers:

We are truly grateful to the four reviewers for the valuable opportunity to revise our manuscript. We have carefully revised the manuscript based on their comments and suggestions. The following is a list of the major changes in this revision:

1. RNA *in situ* hybridization was performed to detect the localization of *GSE9* transcripts in young panicles of rice in detail (lines 128-131; Figs. 2g-h).
2. Additional evidence for the mechanism underlying the role of *GSE9* in grain shape has been added in this revised manuscript (lines 163-178; Supplementary Table 5; Supplementary Fig. 9). The possible mechanism may involve cell expansion and proliferation through amino acid metabolic changes, while the details remain to be fully investigated.
3. More detailed explanations and additional evidence have been provided for the *de novo* origin of *GSE9* gene (lines 202-222 and 297-312; Supplementary Figs. 12-15).
4. More robust analyses for population genetics of *GSE9* were performed, and the relevant results were also modified in the manuscript (lines 245-260; Figs. 5b-d).
5. More *Oryza rufipogon* accessions were included in the analyses of haplotype network and phylogenetic tree for *GSE9* gene (Figs. 5e-f).
6. We added the comparisons of grain shape difference between *GSE9* and *gse9* types within *O. rufipogon* (lines 280-283; Supplementary Fig. 18).
7. Other changes were made based on the comments and suggestions by reviewers, as well as to conform to the format requirements of *Nature Communications*.

Responses to comments by Reviewer #1

The manuscript “A naturally evolved *de novo* gene contributes to rice grain shape difference between *indica* and *japonica*” identified a gene *GSE9* controlling grain shape difference between *indica* and *japonica*. Most *japonica* varieties contain the *GSE9* gene. However, the majority of *indica* varieties absence the start codon. Population evolutionary analyses revealed that *gse9* and *GSE9* were derived from Or-I and Or-III type of wild rice, respectively. *GSE9 de novo* evolved from a previously noncoding region of wild rice *Oryza rufipogon* through the acquisition of start codon. The findings provided new molecular evidence for the *de novo* gene contributes to the genetic and morphological divergence between the two rice cultivated subspecies, and provide a new target for precise manipulation of rice grain shape.

Response: We thank the reviewer for his/her assessment of this manuscript.

Q1. The authors present several data supporting their finding that the key mutation between *GSE9* and *gse9* was a SNP which resulting in the acquiring of start codon. The *GSE9* is only expressed in the *japonica* varieties. In my opinion, it is odd that why this SNP was responsible for the expression difference between *GSE9* and *gse9*? Are there any sequence differences in the promoter region of *GSE9* between *GSE9* and *gse9*. Oddly, despite doing many transgenic rice lines they do not produce a line where only the single SNP is introduced in modern rice, that in accordance with the previous finding would support their hypothesis. I think that would strengthen significantly their paper and would provide a very nice example of how a small change in the DNA can evolve *de novo* gene contributes to rice grain shape difference between *indica* and *japonica*.

Response: Many thanks for the constructive comment. According to your suggestion, we compared the promoter region of *GSE9* between the *japonica* variety Nipponbare (NIP, *GSE9* type) and the *indica* variety Minghui 63 (MH63, *gse9* type), and identified several SNPs and one InDel (Response Figure 1-1a). To further investigate whether these variations lead to the expression difference between *GSE9* and *gse9*, we generated the NIP-*proGSE9*::LUC and MH63-*proGSE9*::LUC constructs, and then performed

transient transactivation assays. Our results showed no significant difference in transcriptional activity between promoter sequences of *GSE9* and *gse9* (Response Figure 1-1b). These findings suggest that the expression level difference between *GSE9* and *gse9* are not caused by the variations in their promoter region. To further investigate whether the A/G variation at the start codon of *GSE9* leads to the expression difference between *GSE9* and *gse9*, we generated NIP-*proGSE9*+A::LUC, NIP-*proGSE9*+G::LUC, MH63-*proGSE9*+G::LUC and MH63-*proGSE9*+A::LUC, respectively. Transient transcription activity assays showed that the sequences of *proGSE9*+A had a significantly higher transcriptional activity than those of *proGSE9*+G and promoter region in NIP and MH63 (Response Figure 1-1b). These results indicated that the variation at the start codon of *GSE9* played an important role in activating the transcription of *GSE9*. We have added these findings into the Results and Methods section of this revised manuscript (lines 202-206 and 470-480).

Given that epigenetic changes could also be responsible for the expression of *GSE9*, we detect the level of DNA methylation in the genomic region of *GSE9*. Our searches against rice methylation database showed that the DNA methylation levels are relatively high in the genomic region of *GSE9* in the *indica* variety MH63, while nearly no DNA methylation can be detected in the *GSE9* genomic region in the *japonica* variety NIP (Response Figures 1-2 and 1-3). In addition, we noticed that the G variation at the start codon site may lead to the high CG methylation level in *indica*, while no DNA methylation level was detected in the start codon of *GSE9* in *japonica*. Further analysis also showed that the degree of DNA methylation varied in the promoter region of *GSE9* between *indica* and *japonica* using a HpaII/MspI digestion-based PCR assay (Response Figure 1-4). The *GSE9* region cannot be digested in all detected *indica* varieties, but was digested in all sampled *japonica* varieties, suggestive of a higher level of DNA methylation in *indica*. These data indicate that different DNA methylation levels may result in the gene expression difference of *GSE9* between *indica* and *japonica*, while the G to A variation at the start codon of *GSE9* also contributes to DNA methylation difference in *indica* and *japonica*. These data and relevant methods were added into the revised manuscript (lines 206-209 and 482-493).

In our initial submission, we revealed that natural variation in the start codon of *GSE9* contributes to grain shape difference between *indica* and *japonica* using the phenotypic and genotypic data of rice re-sequencing varieties. Additional evidence also suggested that the G to A variation at the start codon of *GSE9* contributes to the *de novo* origination of this gene by acquiring the ability of transcription and translation simultaneously. Given that our current analyses have already shown the role of the variation at start codon of *GSE9* gene in phenotypic variation, we did not provide the phenotypic change of a transgenic line with only one SNP at the start codon site of *GSE9* in this manuscript. Nonetheless, we do agree with you that generating a transgenic line with only one single SNP would strengthen our findings that a small change in the DNA can evolve *de novo* gene.

Based on your comments, we have provided an explanation for our major results (lines 202-211 and 306-309).

Response Figure 1-3. Three types of DNA methylation levels of homologous sequences with *GSE9* and its flanking regions in young leaves and panicles of the *indica* variety Minghui 63 (MH63). (a) DNA methylation levels of homologous sequences with *GSE9* and its flanking regions in MH63. Red dotted lines indicate the homologous sequences with *GSE9* genomic region. (b) DNA methylation levels of the homologous sequences with NIP *GSE9* genomic region in MH63. Red dotted lines indicate the homologous sites surrounding the start codon of NIP *GSE9* in MH63. (c) DNA methylation levels of the homologous sites surrounding the start codon of NIP *GSE9* in MH63. Red arrow indicates the G variation at the start codon of *GSE9* in MH63. Red box shows high CG methylation level in the start codon site of *GSE9* in MH63. The data are available from rice methylation database RiceENCODE (Xie et al. *Mol Plant*. 2021, 14(10):1604-1606). Bright blue, orange and magenta boxes indicate the CG, CHG and CHH methylation levels, respectively. Light blue, orange and magenta boxes indicate no CG, CHG and CHH methylation levels.

Response Figure 1-4. HpaII/MspI digestion-based PCR assay showing DNA methylation levels in the promoter region of *GSE9* in *japonica* and *indica*. The sampled *japonica* varieties include Nipponbare (NIP), Zhonghua11 (ZH11), Wuyunjing7 (WYJ7) and Dongjing (DJ). The sampled *indica* varieties include 9311, Minghui63 (MH63), Kasalath, Huanghuazhan (HHZ) and Guangluai4 (GLA4).

Q2. The authors didn't tell us what exactly *GSE9* encodes and how it could affect the cell proliferation and expansion. It is necessary to be addressed for publishing in high impact Journal.

Response: We totally agree with you that it is important to understand the mechanism underlying the role of *GSE9* in regulating grain shape. In this study, we addressed an important but largely unexplored question about the role of *de novo* evolved genes in phenotypic variation of eukaryotes, particularly in crops. We report a *de novo* evolved *GSE9* gene that contributes to the distinguishing trait (i.e., grain shape) difference between *indica* and *japonica*, two most widely cultivated subspecies of rice. Although we are unable to know what exactly *GSE9* encodes thus far because this gene lacks any known sequence characteristics and the homologs of *GSE9* is absent in any known taxa outside *Oryza* species, our transgenic assays, RNA-seq data, and cytological observations suggest that *GSE9* regulates grain shape by affecting both cell expansion and cell proliferation. According to your suggestion, we have provided additional evidence for the potential mechanism underlying the role of *GSE9* in regulating grain shape of rice from the following five aspects:

Firstly, the expression patterns of *GSE9* during panicle development of rice were precisely determined using RNA *in situ* hybridization techniques (Response Figure 1-5). The *GSE9* transcript was observed in young panicle, including anthers, palea and lemma of spikelet hull, suggestive of its role in rice grain development. These findings have been added into the section of Results (lines 128-131), and the Methods section (lines 406-415) in the revised manuscript.

Secondly, decades of plant molecular studies have uncovered several signaling pathways controlling grain size, including the ubiquitin-proteasome pathway, G-protein signaling, MAPK signaling, phytohormone sensing and signaling, and a range of transcriptional regulators (Ren et al. *Sci Bull* (Beijing). 2023, 68(3):314-350). To explore the regulatory relationship between *GSE9* and known functional genes controlling grain shape, we selected a total of 64 well-documented genes regulating grain shape to analyze their transcript level in young panicles of ZH11 and *GSE9* transgenic lines using the RNA-seq data (Response Table 1-1). None of these known grain shape-related gene showed a significantly opposite trend of expression change in ZH11-OE-1 vs ZH11 and ZH11-Cas9-1 vs ZH11 comparisons. Therefore, we speculate that the regulation of *GSE9* in grain shape may not be related to the well-known pathways of grain shape, and possibly represent a novel mechanism. We have added these results into the revised manuscript (lines 163-168).

Response Table 1-1. The expression levels of major regulators of grain shape in young panicle of the wild-type and *GSE9* transgenic plants using the RNA-seq data.

Grain shape signaling				
pathways	Gene name	ZH11-Cas9-1	ZH11	ZH11-OE-1
Ubiquitin-proteasome pathway	GW2	19.03±1.61b	19.25±1.25b	24.34±1.24a
	TUD1	35.61±1.19a	31.19±3.06ab	27.42±3.05b
	OsHRZ2/CLG1	16.07±1.35b	18.99±3.58b	27.9±1.72a
	WTG1/OsOTUB1	72.6±4.94a	65.63±2.47a	63.83±3.28a
	LG1/OsUBP15	28.06±1.48b	31.71±1.3ab	33.81±0.83a
	HDR3	7.15±0.49b	7.98±0.81b	10.37±0.63a
G-protein signaling	RGB1	64.93±1.95a	53.95±1.77b	63.5±0.74a
	RGG1	3.67±0.45a	2.69±0.14b	3.68±0.17b
	RGG2	11.69±0.62a	10.11±0.35ab	9.68±0.67b
	GS3	7.31±0.76a	6.43±0.27a	7.97±1.02a
	RGA1	26.91±1.37b	30.18±3.31ab	37.24±4.2a
	DEP1	6.86±1.02a	6.4±0.97a	6.95±0.92a
MAPK signaling	OsMKKK10	48.82±2b	65.43±2.23a	69.09±1.86a
	OsMKK4	21±2.17ab	22.8±0.47a	17.14±0.95b
	OsMAPK6	26.15±3.79a	35.01±2.53a	35.54±3.76a
	OsMKP1	15.98±0.56a	18.72±0.8a	15.38±1.64a
	OsRac1	29.28±1.45a	25.13±4ab	18.81±0.85b

Phytohormone signaling (Brassinosteroids)	OsERI	27.86±1.86ab	25.28±1.86b	32.65±1.9a
	D11/CPB11	2.76±0.44a	3.36±0.62a	2.47±0.09a
	D2/SMG11	0.37±0.17a	0.45±0.21a	0.41±0.02a
	SLG	126.7±1.36ab	113.41±10.18b	136.82±6.95a
	OsBAK1	36.82±2.25a	37.67±1.48a	42.27±1.8a
	OsBRI1	35.26±2.14a	35.28±1.68a	38.16±2.53a
	OsBSK2	16.61±1.16a	14.54±1.56a	15.22±2.16a
	SG1	3.63±0.53a	3.64±0.6a	2.53±0.08a
	OsBZR1	74.4±11.27a	60.72±6.21ab	45.53±2.55b
	OFP3	4.5±0.42a	4.25±0.38ab	3.28±0.27b
	OsOFP8	0.71±0.11a	0.48±0.16a	0.64±0.04a
	SG2	11.63±0.83a	8.22±1.02b	8.09±1.03b
	GS5	6.12±0.01a	4.05±0.32b	5.17±0.55a
	GSE5	1.39±0.44a	1.04±0.29a	1.15±0.05a
	GSK2	3.1±0.5a	3.17±0.2a	3.58±0.82a
Phytohormone signaling (Auxin)	TSG1/FIB	11.88±1.79a	11.42±1.43a	8.27±0.74a
	OsIAA3	138.25±7.95b	171.12±11.29a	181.37±13.82ab
	OsARF25	39.12±2.12a	36.42±1.97a	37.49±2.65a
	OsARF6	39.28±1.76a	38.14±0.89a	45.04±6.25a
	OsARF4	65.12±1.2b	80.52±2.51a	88.06±7.39a
Phytohormone signaling (Cytokinin)	OsPUP7	15.33±5.47a	6.6±0.89a	15.09±2.31a
	OsPIL15	4.78±0.39a	4.09±0.59a	4.04±1.03a
Phytohormone signaling (Gibberellin)	GW6/OsGASR7	260.19±21.06a	156.19±19.8b	207.15±14.88ab
	OsGASR9	27.62±2.2a	15.19±3.7b	34.22±0.46a
Phytohormone signaling (Ethylene)	OsSPMS1	45.95±0.66a	45.72±2.23a	47.92±1.66a
	OsPAO5	0.01±0.01a	0.01±0.01a	0.01±0.01a
	OsEIN2/MH27	2.6±0.64a	1.87±0.03a	2.09±0.19a
	OsEIL1/MHZ6	9.3±1.61a	9.51±0.23a	8.57±0.42a
	OsEIL2	82.8±5.1a	100.38±16.27a	96.58±6.76a
	OsERF115	0.05±0.04a	0.1±0.14a	0.16±0.06a
Transcriptional regulatory pathway	OsSPL13	7.91±0.85a	1.99±0.52b	0.47±0.29b
	OsSPL16	39.03±3.07a	24.79±3.01b	21.2±1.84b
	GL7/GW7	29.89±2.36a	31.94±3.27a	37.59±3.39a
	OsSPL18	42.11±2.44a	32.01±3.5b	24.94±1.19b
	OsGRF4	4.28±0.35a	3.08±0.34ab	2.45±0.48b
	OsGIF1	0.2±0.06a	0.27±0.11a	0.31±0.16a
	OsLG3	12.4±2.86b	18±0.78a	11.85±0.83b
	OsERF115	0.05±0.04a	0.1±0.14a	0.16±0.06a

OsSNB	16.5±1.18a	15.1±0.5a	12.23±0.96b
SMOS1	10.73±0.52b	10.71±0.96b	14.27±0.43a
An-1	33.21±0.56a	32.33±2.49a	29.56±2.59a
PGL1	0.17±0.08b	0.64±0.08a	0.35±0.12ab
APG	13.3±1.31a	13.97±0.8a	9.61±0.49b
PGL2/OsBUL1	52.42±1.31a	54.45±3a	62.97±6.44a
OsbHLH079	3±0.31a	3.04±0.15a	2.34±0.37a
OsPIL11	0.35±0.14a	0.67±0.1a	0.42±0.22a
OsPIL15/OsbHLH105	4.78±0.39a	4.09±0.59a	4.04±1.03a

Note: Data show means ± SD ($n = 3$) of FPKM values from the RNA-seq data. Different letters indicate significant difference (Tukey's HSD test at $p < 0.05$).

Thirdly, to explore whether the regulation mechanism of *GSE9* in grain shape is related to phytohormone transport or content, we analyzed *GSE9* expression in the wild-type seedlings under the treatment of various phytohormones, and quantified the content of several phytohormones in the young panicles of the wild-type and *GSE9* transgenic lines (Response Figure 1-6). Time-course expression analyses revealed that the *GSE9* transcript level was not significantly induced by the application of all detected phytohormones, including the synthetic cytokinin 6-benzylaminopurine (6-BA), the auxin indole-3-acetic acid (IAA), the gibberellic acid (GA_3), and the precursor of ethylene 1-aminocyclo-propane-1-carboxylic acid (ACC). Among the four determined phytohormones, the content of cytokinin (*trans*-zeatin, *tZ*) showed no significant difference in ZH11 and *GSE9* transgenic plants. The IAA and GA_3 contents were significantly increased in ZH11-OE-1 plants, but showed no significant difference in ZH11-Cas9-1 plants as compared with that in ZH11. In addition, the contents of ACC were significantly decreased in both the ZH11-OE-1 and ZH11-Cas9-1. Given that the expression of *GSE9* gene did not respond to various phytohormones, and the contents of these phytohormones showed no obvious change patterns between *GSE9* transgenic plants and wild-type, we speculate that the regulation of *GSE9* gene in grain shape may not be associated with the changes of phytohormone transport and content.

Fourthly, to further investigate whether the regulation mechanism involves changes in the abundance of other metabolites, we performed untargeted metabolomics analysis. A total of 16 and 10 differential metabolites were detected under ZH11-OE-1 vs ZH11 and ZH11-Cas9-1 vs ZH11 comparisons, respectively (Response Figure 1-7). Notably, among the overlapped metabolites between the two comparisons, the abundance of three metabolites, including L-Glutamine, 4-Hydroxycinnamic acid and 13(S)-HOT, were significantly decreased in ZH11-Cas9-1, but increased in ZH11-OE-1. Given that L-Glutamine and 4-Hydroxycinnamic acid play key roles in amino acid metabolism and that amino acid metabolism is at the foundation of cell proliferation and growth (Zhu et al. *Nat Rev Mol Cell Biol.* 2019, 20:436-450), we speculate that the regulation of *GSE9* in grain shape might involve certain amino acid metabolism during cell proliferation and expansion. In the revised manuscript, we also included the related results into the section of Results (lines 168-172).

Fifthly, to identify the interacting protein of *GSE9*, we carried out yeast two-hybrid (Y2H) assays by screening the cDNA library constructed from the young panicles. A total of 36 positive clones corresponding to 20 annotated genes were identified (Response Table 1-2). However, we noticed that none of these putative interacting proteins were reportedly involved in rice grain shape. In addition, these genes that encode putative interacting proteins did not show significantly opposite trend of expression changes in ZH11-OE-1 vs ZH11 and ZH11-Cas9-1 vs ZH11 comparisons. Therefore, *GSE9* gene might be involved in grain shape through an unrevealed regulatory pathway.

Taken together, these findings indicated that the regulation of *GSE9* in grain shape may not be related to well-known pathways of grain shape. Given that *GSE9* is a *de novo* evolved new gene that lacks any known sequence characteristics and annotated domains, it is likely that *GSE9* might involve a novel mechanism regulating rice grain shape, which affects cell proliferation and expansion likely through changes in amino acid metabolism.

Response Table 1-2. The list of potential interacting proteins of GSE9 and the FPKM values from RNA-seq analysis for these protein-coding genes.

Gene	Annotation	ZH11-Cas9-1	ZH11	ZH11-OE-1
LOC_Os01g41710	Chlorophyll A-B binding protein	3660.96±191.74a	3835.13±607.66a	2253.2±209.88b
LOC_Os01g39330	Basic helix-loop-helix (bHLH) family protein-like	3.02±0.39a	2.91±0.25a	3.22±0.32a
LOC_Os02g10160	Protein of unknown function DUF2048 domain containing protein	11.96±0.43a	11.39±1.08a	12.01±1.37a
LOC_Os02g06640	Similar to polyubiquitin containing 7 ubiquitin monomers	1058.2±15.69b	1487.3±75.16a	1400.68±89.2a
LOC_Os02g43750	Transcription initiation factor TFIID component TAF4 domain containing protein	23.21±0.32b	27.5±2.18ab	31.07±1.36a
LOC_Os03g51600	Tubulin/FtsZ domain containing protein	1357.92±112.16ab	1557.73±76.32a	1243.12±64.06b
LOC_Os03g63480	Ankyrin repeat domain containing protein	21.18±1.27a	18.33±2.12ab	15.07±0.7b
LOC_Os04g57340	Ethylene-responsive transcription factor 7	111.49±16.36a	82.14±23.15ab	62.8±6.33b
LOC_Os04g46140	Transferase family protein	5.94±1.35a	7.07±2.78a	2.68±0.57a
LOC_Os06g02370	Major facilitator superfamily	50.62±2.43a	37.99±6.21a	44.8±4.37a
LOC_Os06g45430	Protein of unknown function DUF284	67.51±2.52a	71.67±3.25a	69.75±3.34a
LOC_Os06g45820	ATP-dependent zinc metalloprotease	48.96±2.57ab	53.97±2.87a	44.18±2.19b
LOC_Os07g08410	Protein of unknown function DUF2346 domain containing protein	54.3±2.53a	52.48±1.58a	49.18±4.64a
LOC_Os08g38060	RING finger protein	57.66±6.31a	49.51±7.14ab	33.51±2.75b
LOC_Os08g38810	BURP domain containing protein	588.86±83.17a	702.54±198.86a	357.6±26.28a
LOC_Os09g36600	Similar to predicted protein	1.73±0.36a	2.86±0.49a	2.11±0.52a
LOC_Os10g41960	Six-bladed beta-propeller	82.74±5.64a	80.88±3.12a	91.66±1.71a
LOC_Os11g09010	Rice ortholog of Arabidopsis phytoalexin deficient 4	27.18±9.53a	35.53±3.82a	27.81±3.52a
LOC_Os12g17600	Similar to Petunia ribulose 1	15.8±3.33a	12.82±3.27a	8.29±1.22a
LOC_Os12g19381	Ribulose bisphosphate carboxylase small chain, chloroplast precursor	2.65±0.76a	3.9±0.45a	3.64±0.33a

Note: Data show means ± SD ($n = 3$) of FPKM values from RNA-seq analysis. Different letters indicate significant difference (Tukey's HSD test at $p < 0.05$).

Q3. In FIG2, the authors found that knock outting *GSE9* in *japonica* varieties can increase the seed length. However, it also decreases the seed width. How about the seed weight of the mutant and wild? It is important for the assessing the application potential of *GSE9*.

Response: In the previous version of the manuscript, we focused on the changes in grain shape of *GSE9* transgenic lines. Thus, the data of seed weight-related traits were only provided in the supplementary materials. Our results showed that the 1,000-grain weight and grain yield per plant decreased in *GSE9* knockout lines, but increased in *GSE9* overexpression lines in *japonica* varieties. Given that overexpression of this gene also results in the increase of grain length and width, manipulating *GSE9* may have great application potential in the improvement of both grain shape and grain yield.

In the revised manuscript, we have added several descriptions on grain weight and yield of *GSE9* transgenic lines, and discussed the potential application on rice production. More detailed information can be found in this revision (lines 123-126 and 327-329).

Q4. The detail gene structure and the DNA and protein sequences are not found in the manuscript. Since *GSE9* is a new gene without full length cDNA, it is better to carry out RACE to confirm the full length of the gene.

Response: Thank you very much for the valuable suggestion. We have added more detailed information on *GSE9* gene structure in Fig. 2a, including the position of 5' terminal, 3' terminal, start codon and stop codon.

The DNA sequence of *GSE9* and its encoded protein were provided in Fig 2c and Fig 4a in our previous version. In the revised manuscript, we changed the figure legend of Fig. 2c into “Comparisons of amino acid sequence encoded by *GSE9* gene in the wild-type ZH11 and the truncated sequence of *GSE9* protein in *GSE9* knockout mutants.” (lines 771-773). The figure legend of Fig 4a was also changed into “Sequence alignment of DNA sequence of *GSE9* gene and its homologous non-coding sequences in various *Oryza* populations” (lines 813-814).

To confirm the full length of the *GSE9* gene, we carried out both 5' and 3' RACE to amplify the cDNA ends of *GSE9* using the specific primers of this gene (Response Figure 1-8). The majority of *GSE9* sequences amplified by RACE are similar to the annotation of the reference genome, but only a total of 900 nucleotides have been amplified for the full-length of *GSE9* gene (Response Figure 1-9). Compared with the annotated full-length sequence of *GSE9* gene, additional 37 nucleotides were amplified at the 3' ends, but 66 nucleotides were not amplified at the 5' ends of this gene. In the revised manuscript, we have added the RACE results into the Results section (lines 111-114). The relevant methods were also provided in the section of Methods (lines 365-375). We also deposited the full-length cDNA of *GSE9* gene in NCBI GenBank with accession number OR050540.1 and added related information in Data Availability (lines 532-533).

Response Figure 1-8. The full-length of *GSE9* gene confirmed by rapid-amplification of cDNA ends (RACE). (a-b) Uncropped gel images for 3' RACE (a) and the alignment of its sequencing results with the annotated sequence (b). (c-d) Uncropped gel images for the first round of 5' RACE (c) and the alignment of its sequencing results with the annotated sequence (d). (e-f) Uncropped gel images for the second round of 5' RACE (e) and the alignment of its sequencing results with the annotated sequence (f). The red arrow indicates the specific products of RACE that are further used for sequencing.

```

1 .....
1 ATATGTAGATCCGACAGAAACACCTCTGATAGTATAAATTTTCATGCCACACAAA
1 .....GAGAAAGAGTAAATACTCCOCTCTCCGAGACCTCGTCTTTCCCTCTCCAC
61 TGACTTGAGAAAGAGTAAATACTCCOCTCTCCGAGACCTCGTCTTTCCCTCTCCAC
95 CATCTCTTCTTCTCTCTCCAACTCTCTGCTCTTCTACTCTCTCCAGCCOCTTACCA
121 CATCTCTTCTTCTCTCTCCAACTCTCTGCTCTTCTACTCTCTCCAGCCOCTTACCA
115 ACGGCTAAGAGTCTACTCTCCATCCCCAATTGCCGAGCCCTGCTTGGACTTGGACC
181 ACGGCTAAGAGTCTACTCTCCATCCCCAATTGCCGAGCCCTGCTTGGACTTGGACC
175 GCGCTGCTCCATCCCCAATGCCGAGCCOCTGATCCGCTCTCCAGAGTCCAGGGAT
241 GCGCTGCTCCATCCCCAATGCCGAGCCOCTGATCCGCTCTCCAGAGTCCAGGGAT
235 ACGGATCTGCTTCCGAGCAGCCTATCCCTCGCTATGCATCACCGCCOCTTGGACC
301 ACGGATCTGCTTCCGAGCAGCCTATCCCTCGCTATGCATCACCGCCOCTTGGACC
295 ACCAAGAGTCACTGCTCCATCCCTCGACTGTGGGCACTGAATCCACATGCCCAACT
361 ACCAAGAGTCACTGCTCCATCCCTCGACTGTGGGCACTGAATCCACATGCCCAACT
355 CGTCCGCCAAGCGCCACCGCCCATCCCTGTCCGTGCGGATCGGATCCGCTCCACA
421 CGTCCGCCAAGCGCCACCGCCCATCCCTGTCCGTGCGGATCGGATCCGCTCCACA
415 AGGTTAGGAGTGCAGATCTGGCCATCGGGGCGCTCTCATCCATCCAGTCTCTGGCCG
481 AGGTTAGGAGTGCAGATCTGGCCATCGGGGCGCTCTCATCCATCCAGTCTCTGGCCG
475 CCACCTCCGATCGTCCGGGAGCGCCACTCCATCCATACACCGGGACACCATCTCTCC
541 CCACCTCCGATCGTCCGGGAGCGCCACTCCATCCATACACCGGGACACCATCTCTCC
535 AATCCCGAGCATGCTCTGCCACTCCGACCCCGCTCCATCTCGAGCACCCGCCACC
601 AATCCCGAGCATGCTCTGCCACTCCGACCCCGCTCCATCTCGAGCACCCGCCACC
595 ATGGCTCCACACCGTGGCTCCTCCAGCTCCTTGGAAITGGAAITGTTTATTGTT
661 ATGGCTCCACACCGTGGCTCCTCCAGCTCCTTGGAAITGGAAITGTTTATTGTT
655 TGTGTTTTATGATTGATTGTTGATATCCAAITGTGGATGCCATGTTCTGATGAGG
721 TGTGTTTTATGATTGATTGTTGATATCCAAITGTGGATGCCATGTTCTGATGAGG
715 TGCTTATGTTGAGAITGGAGTGAITGAGGTGTGTTATGATTGAGATCGGAGAGTAGAT
781 TGCTTATGTTGAGAITGGAGTGAITGAGGTGTGTTATGATTGAGATCGGAGAGTAGAT
775 GACTCAITATAGAGAGAGTGGGTCTTTTCTTACTCTGTTTTTCTCTAGGAAGA
841 GACTCAITATAGAGAGAGTGGGTCTTTTCTTACTCTGTTTTTCTCTAGGAAGA
835 TCTTATCTAGATATTGCTTTTCTCTTTTCTTTTCTCCATGGATCAACTTATTCGACG
901 TCTTATCTAGATATTGCTTTTCTCTTTTCTTTTCTTTTCTTTTCTTTTCTTTTCTTTT
895 TGGCCA
930 .....

```

Response Figure 1-9. Alignment of RACE result (upper line) and annotated sequence (lower line) for the full-length cDNA of *GSE9* gene.

Q5. Phenotypic analyses showed that the *GSE9*-complementary lines had no significant differences in multiple agronomic traits, but more round grains when compared with that of Kasalath. However, in the FIG6, it seems the tiller number is different in the *indica* and *japonica*.

Response: We are so sorry for the mistake. As *GSE9* has no significant effects on other traits including the tiller number, we have modified the cartoon diagram of Fig. 6 in the revised manuscript.

Q6. The Fig3b are not clear and the legend is too simple, which not mentioned the difference between ZH11 and the mutant. In Fig3d it is better to using mm for the Spikelet perimeter.

Response: In the revised manuscript, we have displayed the uncompressed original image of the cross-sections of spikelet hulls in the Fig. 3b to show more clear changes in cell size and cell number between ZH11 and *GSE9* transgenic plants. We also added more description in the legend of Fig. 3 (lines 802-803). In addition, the unit of the spikelet perimeter was changed into mm in the revised Fig. 3d.

Responses to comments by Reviewer #2

In my view, the manuscript 'A naturally evolved *de novo* gene contributes to rice grain shape difference between *indica* and *japonica*' combines two layers of information. The first one is the GWAS-based identification and experimental validation of the *GSE9* gene that contributes to the rice grain morphology. The second layer relates to the claim that this is a '*de novo* gene', and provides some population genetic statistics. The identification and validation of the *GSE9* gene presents some nice experimentation, including transcript analysis, genetic transformation with CRISPR/Cas9, cellular localization of GSE9::GFP in *Nicotiana*, and more.

Response: We agree with the assessment of this reviewer.

Q1. The GWAS analysis identified quite a few peaks beyond the chosen significance threshold. Some of these apparently correspond to genes known to be involved in grain size and shape, and are indicated on Fig. 1a, b. However, these genes are not mentioned in the Figure legend, and it would be appropriate to mention them in the introductory paragraph, too. Furthermore, besides these known genes and the *GSE9* peak, there are several additional signals, and it is not quite clear why the authors focused on the *GSE9* signal. Is this the only signal that is detected for both grain length and width? Even so, why are the authors focusing only on genes that affect length and width simultaneously and disregard signals that affect length and width separately? An explanation of the thought process would be nice.

Response: Thanks for your constructive suggestion. To our knowledge, *GRAIN SIZE 3* (*GS3*) is a major QTL for grain size, and natural variation of this gene contributes to the difference of grain length between *indica* and *japonica* varieties (Fan et al. *Theor Appl Genet.* 2006, 112:1164-1171; Mao et al. *Proc Natl Acad Sci U S A.* 2010, 107:19579-19584). In addition, *GRAIN SIZE ON CHROMOSOME 5* (*GSE5*) is a well-known locus that controls grain width differences between *indica* and *japonica* varieties (Duan et al. *Mol Plant.* 2017, 10:685-694). In this manuscript, we also identified the *GS3* locus for grain length (GL) and *GSE5* locus for grain width (GW) through the GWAS analysis. Given that typical *indica* and *japonica* varieties display obvious

differences in both grain length and grain width, we only focused on the identification of loci simultaneously controlling GL and GW to further understand the genetic basis underlying the grain shape difference between *indica* and *japonica*. Indeed, in our previous analyses, we also found several other signals on chromosome 1, 3 and 9 involving grain length and width simultaneously besides the *GSE9* locus. Given that the peak SNP for GL (chr09:3105176) occurred on chromosome 9, and that the peak SNP for GW on chromosome 9 (chr09:3134839) was also located within a linkage disequilibrium distance centered on the leading SNPs for GL, we then focused on investigating the candidate genes for this locus by expression analyses and transgenic assays. We do agree with the reviewer that more investigations on the other signals are needed to identify more genes controlling grain length or width and to further explore the genetic bases underlying grain shape of rice. We still make efforts on that.

According to your suggestion, we have added more description about the previously reported grain shape-related loci that were also identified in our GWAS analyses, including *GS3*, *GIF1* and *GSE5/GW5*, in the Introduction (lines 74-79), Results (lines 91-92) and Figure Legend (lines 757-758) parts of the revised manuscript.

Q2. A related comment is about a nuanced phrasing of the role of *GSE9*. The abstract says that *GSE9* 'controls' grain shape difference between *indica* and *japonica*. However, the CRISPR/Cas9 experiments showed that the grain length and width changed only about 5% and 3.5%, respectively. Since the median/modal differences between *indica* and *japonica* are much larger (shown on Supplementary fig. 1), it should be clearly formulated that *GSE9* is only one of the genes that contribute to this difference.

Response: We totally agree with you that *GSE9* is only one of the genes that contribute to the grain shape difference between *indica* and *japonica*. To avoid the confusion, we have changed the sentence into the following: “We here report a rice *de novo* gene *GSE9* that contributes to grain shape difference between *indica/xian* and *japonica/geng* varieties” in the Abstract of this revised manuscript (lines 41-42).

Q3. The second layer of the manuscript is a dip into some murky waters. The authors claim that *GSE9* is a '*de novo* gene', i.e. it originated from a non-coding sequence. One fundamental problem with this statement is that depending on how far back in the molecular evolution we go, ultimately all genes originate from non-coding sequences. So what is the definition of a '*de novo* gene'? I thought that perhaps the authors are trying to say that the gene originated in cultivated rice, i.e. after domestication. However, this is still not clear to me, as the text is a bit ambiguous on that. Was the *GSE9* haplotype with the start codon found in wild rice? Parts of the text imply that the start codon was acquired in *O. rufipogon* (lines 41-42, 137-138, 142-144), but the haplotype network does not show wild rice of the *GSE9* type.

Response: Thank you for these insightful comments. We do agree with you that all protein coding genes ultimately evolved from non-coding sequences. In our analysis, we found that the orthologous sequences of *GSE9* gene are present in the syntenic regions of several *Oryza* outgroup species, including *O. barthi*, *O. glumaepatula*, and *O. longistaminata*, but they do not encode an ORF due to absence of the start codon. The full coding sequence of *GSE9* is only present in *O. rufipogon* and most *japonica* varieties. Phylogenetic analyses showed that *japonica* with the *GSE9* type evolved within the group of *O. rufipogon* III (Or-III). These data suggest that *GSE9 de novo* evolved from previously non-coding sequence in *O. rufipogon* and was inherited by *japonica*. We have added this information and other related discussions in the revised manuscript (lines 297-312).

We feel sorry for too few sampled *O. rufipogon* accessions for the haplotype network analysis in our previous submission, which did not clearly show the *GSE9* haplotype with the start codon in common wild rice. To address this issue, a total of 76 *O. rufipogon* accessions were sampled in the revised manuscript after removing the accessions with low sequencing qualities and admixture backgrounds. The revised haplotype network in Fig. 5f has clearly showed that *japonica* varieties with the *GSE9* type was likely derived from Or-III type of *O. rufipogon* with the *GSE9* type.

Q4. The authors have demonstrated that some rice genotypes have the start codon and express *GSE9*, while others lack the start codon and do not express *GSE9*. But how do we know what is the ancestral state? Perhaps the ancestral state is the ATG start codon, and the loss of it is the derived state, in which case we are looking at a pseudogenization rather than a *de novo* gene emergence. Absence of *GSE9* homologs outside of the genus *Oryza* indicates that the gene is relatively young, but it does not tell us whether that particular ATG is derived or ancestral. The haplotype network (Fig. 5f) could be informative in this respect, but is under-interpreted by the authors. The *GSE9* types (I assume it means with the ATG start codon) are scattered all over the network, without showing a single origin. I can think of three possible explanations. (i) the ATG variant emerged multiple times in different haplogroups through convergent evolution (the network suggests at least 6 times, which seems unlikely). (ii) the ATG variant is very old, and the network suffers from poor sampling of wild diversity, which could place the *GSE9* type into all central nodes. (iii) the haplotype network includes recombined sequences, i.e. not true haplotypes. It is not clear to me what region was used for the construction of the network (and the dendrogram). Was it just the gene sequence, or a wider region? I think the genealogy of the haplotypes needs to be analysed more carefully.

Response: Many thanks for your insightful comments. As mentioned above, due to the too few *O. rufipogon* samples, our previous haplotype network and phylogenetic tree did not clearly show the wild rice with the *GSE9* type, and the evolutionary process of the *GSE9* type with the ATG start codon and *gse9* type without the ATG start codon. In the revised manuscript, we have included more *O. rufipogon* accessions to improve the sampling of wild rice diversity. In addition, to avoid the effect of recombined sequences on the haplotypes, we removed the sequences of admixture background shown by population structure of both common wild rice (Wang et al., *Genome Res*, 2017, 27(6):1029-1038.) and cultivated rice (Wang et al., *Nature*, 2018, 557:43–49). A total of 76 *O. rufipogon* accessions and 1,697 cultivated rice varieties were selected for the further analyses. Our comprehensive analyses showed that all AA-genome wild rice relatives that have highly similar sequences to the *GSE9* gene (including *Oryza barthi*,

O. glumaepatula and *O. longistaminata*) and 80% of *O. rufipogon* I (Or-I, the progenitor of *indica*) do not contain the start codon site of *GSE9*, whereas 100% of *O. rufipogon* III (Or-III, the progenitor of *japonica*) contained the start codon site of *GSE9* gene. Thus, the most plausible explanation for these findings is that *gse9* type (i.e., without the start codon) is the ancestral state, while the *GSE9* type (i.e., with the ATG start codon) is the derived state.

To better understand the evolutionary process of *GSE9* gene during the origin and domestication of rice, we constructed the haplotype network and phylogenetic tree in the revised manuscript. Full-length cDNA sequences of *GSE9* gene and their homologs, instead of the coding sequence used previously, from both wild rice and cultivated rice were used in the analyses. The revised haplotype network clearly showed that haplotypes mainly present in *indica* and *japonica* varieties have the closest relationships with Or-I carrying the *gse9* type and Or-III carrying the *GSE9* type, respectively. Several *indica* varieties have the *GSE9* type, likely as a result of gene introgression or independent mutations from *gse9* type. The revised phylogenetic tree also clearly revealed that the clades of *indica* with the *gse9* type and *japonica* with the *GSE9* type fell into those of Or-I and Or-III, respectively. Given that *indica* and *japonica* descended from Or-I and Or-III, respectively (Huang et al., *Nature*, 2012, 490(7421):497-501), we speculated that *indica* carrying the *gse9* type and *japonica* carrying the *GSE9* type were independently derived from Or-I and Or-III.

Taken together, these findings indicate that the *gse9* type without the ATG start codon is most likely the ancestral state, and *GSE9 de novo* originated from a previously non-coding region of wild rice *O. rufipogon* by acquiring a start codon. Population evolutionary analyses further provided evidence for the independent origin of *gse9* in *indica* and *GSE9* in *japonica*. We have added the above information and related discussions in the revised manuscript (lines 189-191 and 269-271). Figs. 5e and 5f were also modified in the revised version.

Q5. Additionally, I would like to see a *Ka/Ks* analysis of the *GSE9* and *gse9* reading frames. If it turns out there is an excess of synonymous substitutions in the *gse9* reading

frame, that would indicate that the reading frame used to be a functional gene before it lost its start codon.

Response: According to your suggestion, we estimated the Ka and Ks values between various *Oryza* species pairs with the *GSE9* and *gse9* type. The results showed that the Ka/Ks ratio ranged from 0.5861 to 1.1810 (Response Table 2-1). These high values indicate the relatively high level of nonsynonymous substitutions in the *GSE9* and *gse9* reading frame, implying that positive selection might have acted on the origin of this *de novo* gene. We further performed a statistical test similar to the McDonald-Kreitman (MK) test to determine the evolutionary divergence between these *Oryza* species with the *GSE9* and *gse9* type (Response Table 2-2). However, the MK test cannot reject the neutral evolution, likely owing to the limited sites within the reading frame of *GSE9* and *gse9*. Nevertheless, we noticed that higher non-synonymous polymorphism was detected in the *GSE9* and *gse9* reading frames. Thus, the most reasonable explanation for these findings is that *GSE9* gene in *Oryza* species may initially experience adaptive evolution driven by positive selection to obtain a new certain function. We have added these findings into the Result section of this revised version (lines 195-199).

Response Table 2-1. Summary of Ka and Ks analysis of the reading frames between *Oryza* species with the *GSE9* and *gse9* types.

Species Pair (GSE9/gse9)	Ka	Ks	Ka/Ks
W3074 (Or-III)/W2302 (Or-I)	0.0131	0.0224	0.5861
W3074 (Or-III)/W2271 (Or-I)	0.0131	0.0111	1.1810
W3074 (Or-III)/ O. barthi	0.0131	0.0224	0.5861
W3074 (Or-III)/ O. longistaminata	0.0537	0.0826	0.6503
W3020 (Or-III)/W2302 (Or-I)	0.0131	0.0111	1.1810
W3020 (Or-III)/W2271 (Or-I)	0.0131	0.0224	0.5861
W3020 (Or-III)/ O. barthi	0.0131	0.0111	1.1810
W3020 (Or-III)/ O. longistaminata	0.0537	0.0702	0.7649

Note: Based on coding sequences of *GSE9* type and the orthologous non-coding sequence of *gse9* type. Ka , nucleotide divergence at nonsynonymous sites; Ks , nucleotide divergence at synonymous sites.

Response Table 2-2. McDonald-Kreitman test of combined *Oryza* species with the *GSE9* and *gse9* types.

Synonymous		Nonsynonymous		G value	Fisher's exact test P (two tailed)
Fixed	Polymorphic	Fixed	Polymorphic		
1	7	1	16	0.304	1.0000 (not significant)

Q6. I have some additional minor comments:

- lines 106-107: '*GSE9* was the most likely candidate gene for *GSE9*' should be corrected to 'LOC_Os09g06719 was the most likely candidate gene for *GSE9*', and similarly in the whole paragraph.

Response: Thank you very much for pointing out this issue. To avoid the confusion, we have changed this sentence into the following: “*GSE9* (LOC_Os09g06719) was the most likely candidate gene for *GSE9* locus (lines 143-144)”. In the previous paragraph, we also added “locus” following the term “*GSE9*” to clearly represent “*GSE9* locus”, and added one sentence “We then designated this gene as *GSE9*.” (lines 107-108).

Q7. - lines 166-170: The two sentences are contradictory. The sequences cannot be simultaneously evolving neutrally and result from natural selection. The low Pi does not necessarily indicate selection (it can have several reasons).

Response: Thank you for pointing out this issue. We do agree with you that the low Pi does not necessarily indicate selection. To better understand the evolution of *GSE9* in various rice populations, we estimated the nucleotide diversity and Tajima’s *D* values for the whole chromosome 9 containing *GSE9* locus in re-sampled wild and cultivated rice populations, respectively (Response Figure 2-2 and 2-3). The relative ratio of nucleotide diversity in *japonica* to wild rice reveals a selective sweep of ~ 1.2 Mb surrounding *GSE9* locus, which exhibited significantly reduced nucleotide diversity in *japonica* compared with wild rice (Response Table 2-3), suggesting a strong artificial selection in *GSE9* locus of *japonica*. The estimated Tajima’s *D* values in *GSE9* locus were significantly negative in *japonica*, implying directional selection across this region. By contrast, no obvious selection was detected in *indica* population because the relative ratio of nucleotide diversity in *indica* to wild rice was higher than that in *japonica* to wild rice in *GSE9* locus, and the Tajima’s *D* value was positive. Taken together, these findings suggested that artificial selection might have contributed to the domestication of *GSE9* in the *japonica* population, which further facilitated the divergence of *indica* and *japonica*. We have added these results and other related information in the revised manuscript (lines 250-260 and 323-326).

Response Table 2-3. The statistical test for the relative ratio of nucleotide diversity of cultivated rice to common wild rice between *GSE9* locus and its flanking genomic regions.

π ratio	GSE9 locus	GSE9 flanking regions	P value
Oryza sativa japonica/Oryza rufipogon	0.0558	0.1503	1.17e-19
Oryza sativa indica/Oryza rufipogon	0.7686	0.6381	1.00e-10

Note: Statistical analysis was performed by one sample *t*-test.

Q8. - lines 210-211: It should be rephrased. The text does not really provide a strategy.

Peter Civan, 24 Jan 2023

Response: We are sorry for the misleading. We have changed this sentence into the following: “Importantly, we provide a new gene target for the improvement of grain shape in rice” in the revised version (lines 331-332).

Responses to comments by Reviewer #3

In this manuscript, Chen and co-authors describe a new rice *de novo* gene, *GSE9*, which encode a functional protein in the variety *Oryza sativa japonica/geng*, whereas the orthologous sequence *gse9* is not expressed in *Oryza sativa indica/xian* and other closely related *Oryza* species. Through a series of thorough experimental assays, the authors demonstrate that *GSE9* is involved in determining the grain shape differences between the *japonica* (ovate grains) and *indica* (slender grains) varieties. Comparative genomic analyses performed here showed that *GSE9* is characterized by an enabling mutation that established a new start codon. Most *japonica* varieties contain this start codon, as opposed to a very few *indica* varieties. Using population and comparative genetic data the authors suggest that *GSE9* originated from the type OR-III of the wild rice *Oryza rufipogon*.

Increasingly, *de novo* gene birth is described as one of the most fundamental and common processes that originate novel genes across eukaryotes. Although *de novo* genes are heralded as potentially major contributors to phenotypic variation and evolutionary innovations, the functional impact of *de novo* genes has been thoroughly assessed only in a limited number of cases.

This work represents one of the most comprehensive analysis of the functional role of *de novo* genes ever conducted, and the most significant contribution to dissecting the origin and phenotypic impact of *de novo* genes in monocots, with a comparable importance to the only other well characterized *de novo* gene in plants, the *Arabidopsis thaliana*-specific gene *QQS*.

The possible application of this discovery to rice grain shape manipulations is noteworthy.

Response: We thank the reviewer for the above assessment.

Q1. The methodological approach is sound and the data presented are convincing, although not comprehensive, as detailed below.

My major concern about the paper is about the association between the emergence of the start codon that characterize *GSE9* with the expression and functionality of *GSE9*

itself. It is my opinion that the body of evidence provided in the manuscript does not rule out the possibility that a functional GSE9 protein evolved before the formation of the canonical start codon that is common in *O. sativa japonica*.

According to the sequences shown in fig 4a, most of the open reading frame of *GSE9* is also present in the orthologs from the other species except in *O. glumipatula* and *O. rufipogon* (W0106), which share a stop codon around the middle of the ORF (CGA to TGA). *O. rufipogon* (W1943), *O. sativa indica* (9311), *O. barthi* and *O. longistaminata* orthologs to *GSE9* encode for a putative protein with a start codon at position 85 of the *GSE9* codon (MQIWP..), as a quick BLAST search confirmed. These putative proteins are very similar to the GSE9 protein and include the region that contains the transmembrane domain/alpha-helix shown in fig. 4d. This suggests that a certain degree of conservation of the GSE9 protein occurs across multiple *Oryza* species, which share the ability to encode a shorter, but possibly functional, protein.

Because *gse9* is not transcribed in the panicles of non-*japonica* *Oryza* species and varieties (fig. 4e), there is no indication whether proteins encoded by *gse9* genes are indeed functional.

This does not contradict the main conclusions of the manuscript. However, it suggests that the evolutionary history of *GSE9* might be more complicated than proposed in the article. In particular, it is possible that the G to A change that established the *GSE9* gene start codon was not instrumental to encode a functional protein. A potentially more impactful change seems to be the transition from a non-transcriptional (or very low transcriptional) state to a transcribed coding sequence that occurred in *O. sativa japonica*, and to a much smaller extent in *O. rufipogon*, according to fig. 4e.

Response: Many thanks for your detailed and insightful comments. We also noticed that there are at least three putative start codons in the open reading frame (ORF) of *GSE9*, which are located at position 1-3, 82-84 and 253-255 of the *GSE9* sequence, respectively. All the corresponding putative proteins include the region that contains the transmembrane domain/alpha-helix, suggesting that *GSE9* gene may stepwise evolve from a previously shorter ORF. To further determine the protein products of these putative ORFs, we performed extensive searches against proteomic database, and

identified one specific peptide with 27 amino acids for GSE9 protein (ProteomeXchange accession: PXD001046). Notably, the sequence encoding the specific peptide starts from the start codon at position 76 of the *GSE9* codon before the location of second and third putative start codon site. Thus, the most likely translation initiation site of GSE9 is the canonical start codon site that is common in *O. sativa japonica*.

To investigate the association between the A/G variation at the start codon of *GSE9* and its transcriptional ability, we performed transient transcription activity assays for the *GSE9* promoter region, *proGSE9+A* and *proGSE9+G* in the *japonica* variety Nipponbare (NIP, *GSE9* type) and the *indica* variety Minghui 63 (MH63, *gse9* type) (Response Figure 3-1). Our results showed no significant difference in transcriptional activity between the promoter sequences of NIP and MH63, indicating that the expression difference between *GSE9* and *gse9* may not result from the variations in the promoter region of *GSE9*. Notably, the sequences of *proGSE9+A* had significantly higher transcriptional activity than those of *proGSE9+G* and *proGSE9* in NIP and MH63. Additional data revealed that different DNA methylation levels may result in the gene expression difference of *GSE9* between *indica* and *japonica*, while the G to A variation at the start codon of *GSE9* also contributes to DNA methylation difference in *indica* and *japonica* (Response Figures 3-2 and 3-3). These findings indicated that the G to A variation at the start codon of *GSE9* plays an important role in activating the transcription of *GSE9*, which may lead to the acquisition of transcription and translation of this gene simultaneously. Our results might provide an example for the simultaneous ORF transcription model of *de novo* origination (Zhang et al. *Nat Ecol Evol.* 2019, 3:679-690).

According to your suggestion, we have added additional explanations and discussions on the evolutionary history of *GSE9* in the revised manuscript (lines 202-215 and 306-309).

Q2. Additional data do not provide support to the claim that the *GSE9* new start codon is responsible for the acquisition of functionality by the *GSE9* protein.

For instance, the Extended Data Fig. 3a shows that the *O. indica* varieties with the *GSE9* gene type, but not the *gse9* type, are transcribed. The authors suggest that “These findings indicated that natural variation in the start codon site of *GSE9* has large effects on the grain shape of rice, which can be further exploited for the improvement of grain

shape.” However, this can also be explained by regulatory changes linked to the start codon site of *GSE9*.

Indeed, the complementary lines of Kasalath were built using a construct that contains the putative promoter of *GSE9*. Lines 250-252: “To generate the complementation lines of *GSE9* gene in *indica* background, the coding sequence and 2-kb promoter region was amplified and cloned into the construct pCAMBIA1301.”

It is therefore possible that the regulatory changes responsible for the expression of the *GSE9* are more functionally important than the G to A mutation establishing the start codon to produce a functional protein by enabling the transcription of this gene.

Response: Thanks for your insightful comments. As mentioned above, the canonical start codon site of *GSE9* gene established by G to A mutation that is common in *japonica* rice is the most likely translation initiation site of GSE9 protein. Further analyses revealed that the G to A variation at the start codon of *GSE9* plays an important role in activating the transcription of *GSE9*, which may lead to the acquisition of transcription and translation of this gene simultaneously. These findings also support the simultaneous ORF transcription model of *de novo* origination (Zhang et al. *Nat Ecol Evol.* 2019, 3:679-690). Thus, we speculate that the G to A mutation at the start codon of *GSE9* may contribute equally to the transcription and translation of this gene.

In our initial submission, we revealed that the natural variation in the start codon site of *GSE9* gene had large effects on grain shape based on the phenotypic difference between cultivated rice varieties with the *GSE9* and *gse9* types. In addition, introgression of the genomic region of this gene lead to the changes of grain shape in *indica* variety without the start codon of *GSE9*. According to your suggestion, we observed the phenotypic changes in various *O. rufipogon* accessions, and found that accessions with the *gse9* type (without ATG) had more slender grains than those with the *GSE9* type (with ATG). All these findings suggest that the G to A mutation might have contributed to the acquisition of certain function of this *de novo* gene in *Oryza* species.

Taken together, the most plausible explanation for these findings is that *GSE9* may obtain the functional elements required to produce both messenger RNA (mRNA) and

protein by establishing the start codon through G to A variation. We have added the above information and related discussions in this revised version (lines 202-215 and 306-309).

Q3. The authors suggest that “*indica* carrying the *gse9* type and *japonica* carrying the *GSE9* type were derived from Or-I and Or-III, respectively.”, based on figs. 5e and 5f phylogenetic and haplotype data. I find this interpretation debatable. There are too few *O. rufipogon* samples to very clearly address the question of where the *GSE9* type evolved. Looking at Fig. 5e., I don’t see “that the clades of *indica* carrying the *gse9* type and *japonica* carrying the *GSE9* type nested within those of Or-I and Or-III, respectively (Fig. 5e).” This could be due to the quality of the figure, but it appears to be primarily an issue with undersampling of *O. rufipogon*. Furthermore, the phylogeny shows no statistical support values for potentially key nodes supporting the ‘nesting’ of clades. In fig. 5f, some Or-III haplotypes are more closely related to *indica* than *japonica*, but it’s not stated if these *indica* haplotypes contain *GSE9* or *gse9*.

Response: Many thanks for pointing out this issue. Due to too few *O. rufipogon* samples used in our previous submission, the phylogenetic and haplotype data did not clearly show the wild rice carrying the *GSE9* type and the evolutionary process of the *GSE9* type with the ATG start codon. To better illustrate the evolutionary process of *GSE9* type, a total of 76 *O. rufipogon* accessions were re-sampled after removing the accessions with low sequencing qualities and admixture backgrounds. The bootstrap supporting values for the phylogeny were estimated using a total of 1000 nonparametric bootstrap samplings. The revised phylogenetic tree clearly showed that the clades of *indica* carrying the *gse9* type and *japonica* carrying the *GSE9* type nested within Or-I and Or-III, respectively. In addition, the revised haplotype network clearly showed the haplotypes mainly present in *indica* and *japonica* varieties have the closest relationships with Or-I carrying the *gse9* type and Or-III carrying the *GSE9* type, respectively. Several *indica* varieties have the *GSE9* type, likely as a result of gene introgression or independent mutations from *gse9* type. Given that *indica* and *japonica* have been suggested to be descended from Or-I and Or-III, respectively (Huang et al.,

Nature, 2012, 490(7421):497-501), we speculate that *indica* carrying the *gse9* type and *japonica* carrying the *GSE9* type were independently derived from Or-I and Or-III type of *O. rufipogon*.

In this revised manuscript, we have modified Figs. 5e and 5f. Given that it is hard to show bootstrap supporting values in the circle phylogeny of Fig. 5e, we provided a traditional rectangular tree with support values in Supplementary Fig. 17, in which the key nodes that support the nesting of clade were also magnified to display.

Q4. Fig. 6. This figure could be inaccurate if experiments show that *GSE9* expression and protein functionality do not depend on the establishment of the new start codon. Also, expression in *O. rufipogon* is much lower—barely present, really—than in *japonica*.

Response: Thank you for these comments. In our initial submission, Fig. 6 illustrated the stepwise origination processes for the *de novo* gene *GSE9*. As we mentioned above, the G to A variation at the start codon of *GSE9* contributes to the *de novo* origination of this gene by acquiring the ability of transcription and translation simultaneously. Given that *GSE9* gene may obtain the functional elements required to produce both messenger RNA (mRNA) and protein by establishing the start codon through G to A variation, we then used green bars and black arrows to indicate the start codon and detected expression signal of this *de novo* gene, respectively. In this revised manuscript, we modified Fig. 6 in order to distinguish the different expression levels of *GSE9* gene in *O. rufipogon* and *O. sativa japonica*. We used the solid arrow to indicate the stronger transcriptional signal of *GSE9* gene in *japonica* rice, and the dotted arrow to indicate the relatively low transcriptional level of this gene in *O. rufipogon* Or-III. In addition, a purple bar was added into the orange box in *O. glumaepatula* to indicate the premature stop codon present in the orthologous sequences of this gene in *O. glumaepatula*. The figure legends of Fig. 6 have also been modified (lines 854-859).

Q5. Furthermore, it would be important to indicate if grain shape in Or-I and Or-III mirror the shape of *japonica* and *indica*, respectively.

Response: We totally agree with you that it would be important to investigate whether the grain shape difference between Or-I and Or-III type of wild rice mirror that of *indica* and *japonica*. In the revised manuscript, Prof Xiaoming Zheng have provided the genotype and grain shape phenotype of 95 *O. rufipogon* accessions. Based on the A/G variation at the start codon of *GSE9*, 51 of 95 *O. rufipogon* accessions were classified as the *GSE9* type and 44 as the *gse9* type. Further analyses showed that common wild rice with the *gse9* type displayed longer and narrower grains than that with the *GSE9* type (Response Figure 3-4), consistent with the typical grain shape in cultivated rice of the *gse9* and *GSE9* types, respectively. Given that most of Or-III carried the start codon of *GSE9*, but the majority of Or-I carried the G variation at the start codon, Or-I and Or-III can be considered as common wild rice with the *gse9* type and *GSE9* type, respectively. Thus, these findings revealed that the grain shape divergence may start in common wild rice with the *gse9* and *GSE9* types, and was then inherited by *indica* and *japonica*, respectively. We have added these findings into this revised manuscript (lines 280-283 and 320-322).

Q6. Suggested improvements. Given the consideration above, the following tests/experiments are recommended. A construct containing the 2kb of the promoter region but without the start codon found in most *japonica* varieties should be tested in a *japonica* line, and phenotypic data collected to show if the new start codon is indeed required to produce the observed phenotype of the wild-type *GSE9* gene.

Response: According to your suggestion, we compared the grain shape of *GSE9pro::GUS* transgenic plants that carried the 2kb promoter region but do not contain the start codon of *GSE9* gene, with that of the wild-type Zhonghua11 (a typical *japonica* variety). The phenotypic data showed no significant difference in grain length and grain width between *GSE9pro::GUS* transgenic plants and the wild-type Zhonghua11 (Response Figure 3-5). These findings suggest that the new start codon establishment of *GSE9* gene is essential to contribute to the grain shape changes of rice. We have added these findings in this revised manuscript (lines 131-135).

Q7. Additional comparative genomic analyses should be presented to show if the promoter region of the *GSE9* gene contains unique mutations compared to the *gse9* orthologs, and if so, the mutated regions should be also tested *via* one or multiple constructs to verify their possible role in enabling the transcription of *GSE9*. Epigenetic changes could also be responsible for the expression of *GSE9* and need to be discussed in the paper.

Response: Thanks again for the above suggestions. As indicated above, we performed a comparative analysis of the promoter region of the *GSE9* gene with the *gse9* orthologs, and detected several variations between the two types (see above Response Figure 3-1a). However, further transient transcription activity assays showed that the 2-kb promoter sequence of the *japonica* variety Nipponbare (NIP, *GSE9* type) displayed no significant difference in transcriptional activity with that of the *indica* variety Minghui 63 (MH63, *gse9* type) (see above Response Figure 3-1b), suggesting that the expression difference between *GSE9* and *gse9* types does not result from the variations in the promoter region of *GSE9*. By contrast, the sequences of *proGSE9*+A had significantly higher transcriptional activity than those of *proGSE9*+G and *proGSE9* in NIP and MH63. These findings suggest that the G to A variation at the start codon of *GSE9* plays an important role in activating the transcription of *GSE9*.

Given that epigenetic changes could also be responsible for the expression of *GSE9*, we detect the level of DNA methylation in the genomic region of *GSE9*. Our searches against rice methylation database showed that the DNA methylation levels are relatively high in the genomic region of *GSE9* in the *indica* variety MH63, while nearly no DNA methylation can be detected in the *GSE9* genomic region in the *japonica* variety NIP (see above Response Figures 3-2 and 3-3). In addition, we noticed that the G variation at the start codon site may lead to the high CG methylation level in *indica*, while no DNA methylation level was detected in the start codon of *GSE9* in *japonica* (see above Response Figures 3-2 and 3-3). Further analysis also showed that the degree of DNA methylation varied in the promoter region of *GSE9* between *indica* and *japonica* using a HpaII/MspI digestion-based PCR assay (Response Figure 3-6). The *GSE9* region cannot be digested in all detected *indica* varieties, but was digested in all sampled *japonica* varieties, suggestive of a higher level of DNA methylation in *indica*. These data indicate that the difference of DNA methylation levels may result in the gene expression difference of *GSE9* between *indica* and *japonica*, while the G to A variation at the start codon of *GSE9* is important for DNA methylation difference in *indica* and *japonica*.

Taken together, the most plausible explanation for these findings is that the G to A variation at the start codon of *GSE9* activates the transcription of *GSE9*. We have added these results into the Result section (lines 202-211).

Response Figure 3-6. HpaII/MspI digestion-based PCR assay showing DNA methylation levels in the promoter region of *GSE9* in *japonica* and *indica*. The sampled *japonica* varieties include Nipponbare (NIP), Zhonghua11 (ZH11), Wuyunjing7 (WYJ7) and Dongjing (DJ). The sampled *indica* varieties include 9311, Minghui63 (MH63), Kasalath, Huanghuazhan (HHZ) and Guangluai4 (GLA4). The *GSE9* region cannot be digested in all detected *indica* varieties, but was digested in all sampled *japonica* varieties, suggestive of a higher level of DNA methylation in *indica*.

Q8. Data from additional *O. rufipogon* varieties should be included, and if not available this should be discussed in terms of the possible scenarios of *GSE9* evolution.

Response: In this revised manuscript, we have included more *O. rufipogon* accessions to construct the haplotype network and phylogenetic tree to better understand the evolutionary process of *GSE9* gene. The results clearly showed that *indica* carrying the *gse9* type and *japonica* carrying the *GSE9* type were independently derived from Or-I and Or-III type of wild rice. More detailed information can be found in Figs. 5e-f of the revised manuscript.

Q9. Data on *O. rufipogon* grain shape for the varieties Or-I and Or-III should be added.

Response: We have obtained the genotype of 95 *O. rufipogon* accessions, and compared the grain shape between common wild rice of the *GSE9* type and that of the *gse9* type. Our results showed that common wild rice of the *gse9* type displayed slender grains than that of the *GSE9* type (see above Response Figure 3-4). The relevant results and discussions have been added in this revised manuscript (lines 280-283 and 320-322).

Q10. Minor comments.

Title. I am not sure what the authors mean by ‘A naturally evolved’. It may be more appropriate to indicate the time of origin of this gene, i.e. ‘*Oryza sativa*-specific’.

Response: Thanks for this comment. In this study, we revealed that *GSE9* is a protein-coding functional gene that *de novo* evolved from previously non-coding sequence of *O. rufipogon*, and was then inherited by most of *japonica* varieties. Because the evolution of *O. rufipogon* was unlikely to be accompanied by artificial selection, we speculate that *GSE9* gene might have undergone natural selection during the origination process in *O. rufipogon*. Therefore, despite the fact that *GSE9* is an *Oryza*-specific gene, using “A naturally evolved” may be more appropriate to show the origination and evolutionary characteristic of this *de novo* evolved gene.

Q11. Line 56: ‘forece’ should be ‘force’

Response: We are sorry for the mistake. We have revised it. Thank you very much!

Q12. Line 57. ‘modification of existing genes’ is not a process of new gene evolution. Modification implies changes of the current status, i.e. sequence, of a gene. New but non-*de novo* genes evolve *via* duplication, TE exaptation, chimerism, horizontal gene transfer and other processes that are not quite ‘modifications’, except maybe chimerism. This should be rephrased.

Response: We totally agree with you that several mechanism underlying the origin of new genes, including duplication, TE exaptation, chimerism, horizontal gene transfer and other processes, are not quite “modification of pre-existing genes”. However, all of them involve recruitment of pre-existing genes or parts of them into new functions. By contrast, the *de novo* evolved genes are novel genes that arise from previously noncoding DNA (i.e., they are not derived from pre-existing protein-coding sequences). In the revised manuscript, we have changed this sentence into the following: “As a driving force of evolutionary innovation, new genes can evolve from pre-existing genes or gene fragments, including gene duplication, horizontal gene transfer, gene fusion and other mechanisms. By contrast, novel protein-coding genes may also originate *de novo* from previously non-coding sequences.” (lines 57-60).

Q13. Reference 2 seems misplaced or not appropriate, there is no mention of *de novo* genes and only one reference about gene duplication, while it contains an extensive description of gene regulatory network changes influencing evolution. However, these changes have nothing to do with the emergence of new genes. Reference 3 is a recent excellent review of *de novo* genes that should be cited in Line 57.

Response: Thanks for pointing out the citation issue. In this revision, we have deleted the reference 2 and kept one reference (Long et al., *Nat Rev Genet*, 2003, 4: 865-875) for the mechanisms underlying the origin of new genes (line 59). In addition, the excellent review of *de novo* genes (Carvunis et al., *Nature*, 2012, 487: 370-374) has been cited for the sentence on *de novo* gene birth (line 60).

Q14. Lines 160-63. Given the low number of locations, the association between latitude and *GSE9/gse9* is inconclusive. Also, analyses based on climate variables and niche reconstructions are better suited to investigate this aspect than crude latitudinal data.

Response: Thanks for the above suggestions. We agree with you that the low number of locations may lead to the inconclusive association between *GSE9/gse9* and their latitude distribution. Indeed, during the preparation of our initial submission, we tried to display the 72 country locations of all rice samples in the geographic map. However, it was hard to show clearly the association between the *GSE9/gse9* and their latitude distribution due to the too many locations. Therefore, we only investigated the association between the *GSE9/gse9* and their latitude distribution in nine continents, which led to a plausible conclusion that the proportion of *GSE9* type has a trend of increasing in higher latitudes.

In the revised manuscript, we tried to perform analyses based on climate variables and niche reconstructions. Because only country and continent information for the rice accessions can be obtained from 3k rice genomes project, we were unable to perform the analysis of niche reconstructions in the absence of detailed latitude and longitude information for each sample. Thus, we only added the terrestrial temperature averages from 1970 to 2000 A.D., obtained from WorldClim (Fick and Hijmans. *Int J Climatol*,

2017, 37 (12): 4302-4315), into the revised geographic map (Response Figure 3-7). In addition, only eight continent locations from re-sampled cultivated rice were marked on the geographic map because a total of 618 accessions with admixture backgrounds, shown by population structure, were removed from this analysis. More detailed information can be found in Results (lines 241-242), Methods (lines 501-502), and Fig. 5a of the revised version.

Response Figure 3-7. Geographic distributions of 1,697 cultivated rice varieties. Blue and red circles indicate the *GSE9* and *gse9* type, respectively.

Q15. Fig. 5. The colors of *GSE9* and *gse9* appear to be switched between 5a and 5e.

Response: In this revised manuscript, the colors of *GSE9* and *gse9* in Fig. 5a have been changed to be consistent with those in Fig. 5e.

Q16. In Figure 5e I found it difficult to point to where Or-I and Or-III are in the tree, these specimens should be better highlighted by different colors or labels on the tree.

Response: In the revised manuscript, we have sampled more *O. rufipogon* accessions carrying the Or-I or Or-III type to construct the phylogenetic tree. The more distinguishable colors were also used to label Or-I and Or-III in the revised tree.

Responses to comments by Reviewer #4

Comments on “A naturally evolved *de novo* gene contributes to rice grain shape difference between *indica* and *japonica*” by Rujia Chen et al.

The authors inferred a *de-novo* gene associated with rice grain shape in *japonica* by GWAS analysis. The knock-out analyses candidate genes perfectly showed the expected results. Therefore, I think that this manuscript is deserved for publication in a great journal. It is my great honor to review this manuscript. However, I think that this manuscript contains several minor problems of evolutionary analyses. Therefore, I recommend that the authors improve evolutionary analyses.

Response: We are truly grateful to this reviewer for the above comments.

The followings are my comments:

Major comments

Q1. This manuscript did not perform robust analyses in population genetics. In Fig. 5b, the authors did not see any statistical test to identify the difference *japonica* and *indica* subspecies. Also, the authors should infer such the values in genomic scale. The values inferred from genomic scale can be values assuming neutral evolution. Please compare the values of *GSE9* flanking regions with those assuming neutral evolution.

Response: Thank you for pointing out these issues. In the revised manuscript, a number of cultivated rice varieties with admixture backgrounds have been removed in our population genetic analyses, and we have re-analyzed five genetic divergence parameters between *indica* and *japonica* for *GSE9* and its flanking regions. Similar to our previous analyses, the estimated values in *GSE9* locus are relatively high compared to its flanking genomic regions (Response Figure 4-1). Further one sample *t*-test showed that the estimated values for *GSE9* locus are significantly different from those of its flanking regions (Response Table 4-1), suggesting a relatively high level of genetic differentiation in *GSE9* locus between two rice subspecies. The average *F_{ST}* values in each 100-kb window were also estimated at the whole-genome level between *indica* and *japonica* (Response Figure 4-2). The results revealed high divergence in the region of *GSE9* locus (top 10% of *F_{ST}* values). We have added these results into this

revised manuscript (lines 245-248). The relevant methods were also added into the revised version (504-506). Fig. 5b was also modified in the revised version.

To further determine whether directional selection has contributed to the origin and domestication of *GSE9*, we estimated the nucleotide diversity and Tajima's *D* values in the whole chromosome 9 in re-sampled wild and cultivated rice populations, respectively. The relative ratio of nucleotide diversity in *japonica* to wild rice reveals a selective sweep of ~ 1.2 Mb surrounding *GSE9* locus, which exhibited significantly reduced nucleotide diversity in *japonica* compared with wild rice (Response Figure 4-3 and Response Table 4-2), suggesting a strong artificial selection in *GSE9* locus of *japonica*. The estimated Tajima's *D* values in *GSE9* locus were significantly negative in *japonica* (Response Figure 4-4), implying directional selection across this region. By contrast, no obvious selection was detected in *indica* population because the relative ratio of nucleotide diversity in *indica* to wild rice was higher than that in *japonica* to wild rice in *GSE9* locus, and the Tajima's *D* value was positive (Response Figures 4-3 and 4-4; Response Table 4-2). Taken together, these findings show that artificial selection might have contributed to the domestication of *GSE9* in the *japonica* population, which further facilitated the divergence between *indica* and *japonica*. We have added these findings and related information in this revised manuscript (lines 250-260 and 322-326). Fig. 5c and Fig. 5d were also modified in the revised version.

Response Figure 4-1. The estimated parameters of genetic differentiation between *japonica* and *indica* for *GSE9* locus and its flanking genomic regions. Red box indicates the *GSE9* locus.

Response Table 4-1. The statistical test for the estimated parameters of genetic differentiation between *japonica* and *indica* for *GSE9* locus and its flanking genomic regions.

Parameters	GSE9 locus	GSE9 flanking regions	P value
Haplotype F_{ST}	0.36	0.12	1.80e-11
Nucleotide F_{ST}	0.86	0.71	3.03e-05
Nei's G_{ST}	0.22	0.07	2.91e-12
Hudson's G_{ST}	0.20	0.06	3.92e-12
Hudson's H_{ST}	0.20	0.06	3.93e-12

Note: Statistical analysis was performed by one sample *t*-test.

Response Figure 4-2. Plots of the population-differentiation statistic (F_{ST}) between *indica* and *japonica* across 12 rice chromosomes. The F_{ST} was estimated in each 100-kb window and the horizontal line indicates the threshold value of top 10% of F_{ST} at the whole-genome level between *indica* and *japonica*.

Response Table 4-2. The statistical test for the relative ratio of nucleotide diversity of cultivated rice to common wild rice between *GSE9* locus and its flanking genomic regions.

π ratio	GSE9 locus	GSE9 flanking regions	P value
Oryza sativa japonica / Oryza rufipogon	0.0558	0.1503	1.17e-19
Oryza sativa indica / Oryza rufipogon	0.7686	0.6381	1.00e-10

Note: Statistical analysis was performed by one sample *t*-test.

Q2. In Fig. 5c, I cannot understand the meaning of “low nucleotide diversity (π) for the *GSE9* locus in both cultivated and wild rice might be the result of natural selection”. Again, the authors did not perform any statistical test. I think that the authors would like to identify selective sweep in *japonica* but no selective sweep in *indica*. If so, the author can infer difference of nucleotide diversity between *japonica* and *indica* in *GSE9*

locus. The difference of nucleotide diversity can be compared with the nucleotide diversity in neutral evolution. The nucleotide diversity of neutral evolution can be inferred from genomic scale analysis. I think that you can do the similar analyses in a *Genome Research* paper (Shirai et al., Positive selective sweeps of epigenetic mutations regulating specialized metabolites in plants. 2021). The authors can try to compare nucleotide diversity in population either with or without ATG in *GSE9*. I'm curious to see selective sweep in *japonica*.

Response: Many thanks for these insightful comments and suggestions. We agree with the reviewer that positive selective sweep in *japonica* and but no selective sweep in *indica* is important for the divergence of this locus between *indica* and *japonica*. In our analysis, the distribution of A/G variations at the start codon site of *GSE9* was unbalanced in *indica* and *japonica*. The vast majority of *indica* varieties (1028 out of 1051, 97.81%) do not possess the start codon at the *GSE9* locus due to the single nucleotide variation of A to G, whereas most of *japonica* varieties (640 out of 646, 99.07%) have the start codon site ATG in *GSE9*. Thus, we defined the rice varieties with ATG and without ATG in the locus as *GSE9* and *gse9* types, respectively. Given that *japonica* varieties with *gse9* type (without ATG) and *indica* varieties with *GSE9* type (with ATG) was too few to accurately estimate the relevant parameters, we then estimated the nucleotide diversity in the whole *indica* and *japonica* populations. The relative ratio of nucleotide diversity in cultivated rice to wild rice was then calculated to determine the selective sweep region. We noticed that a selective sweep of ~ 1.2 Mb surrounding *GSE9* locus exhibited significantly reduced nucleotide diversity in *japonica* compared with wild rice, suggestive of a strong artificial selection in *GSE9* locus of *japonica*. By contrast, the relative ratio of nucleotide diversity in *indica* to wild rice was higher than that in *japonica* to wild rice in *GSE9* locus (see above Response Figure 4-3 and Response Table 4-2). Further neutral evolution analysis showed that the estimated Tajima's *D* values for the *GSE9* locus are significantly negative in *japonica*, but positive in *indica* (see above Response Figure 4-4). These findings revealed that artificial selection might have contributed to the domestication of *GSE9* in the *japonica* population, which further facilitated the divergence between *indica* and *japonica*. We

have added these findings and related information in this revised manuscript (lines 250-260 and 322-326). Fig. 5c was also modified in the revised version.

3) In Fig. 5d, the authors described that “none of Tajima’s D values for all estimated populations were statistically significant (Fig. 5d), implying that *GSE9* locus did not escape from neutral evolution during rice domestication.” I think that the authors should infer Tajima’s D values in *japonica* with ATG, *japonica* without ATG, *indica* with ATG and *indica* without ATG. The authors then examine the differences of Tajima’s D values among different populations.

Response: As mentioned above, the vast majority of *indica* varieties do not possess the start codon at the *GSE9* locus, whereas most of *japonica* varieties have the start codon site ATG in *GSE9*. We then estimated the nucleotide diversity and Tajima’s D value for the whole *japonica* and *indica* populations. The estimated Tajima’s D values for the *GSE9* locus are significantly negative in *japonica* (see above Response Figure 4-4), suggestive of a strong directional selection. The relative ratio of nucleotide diversity in *japonica* to wild rice revealed a selective sweep of ~ 1.2 Mb surrounding *GSE9* locus (see above Response Figure 4-3), implying a strong artificial selection in *GSE9* locus of *japonica*. By contrast, the Tajima’s D values for the *GSE9* locus are significantly positive in *indica*, and the relative ratio of nucleotide diversity in *indica* to wild rice was higher than that in *japonica* to wild rice (see above Response Figures 4-3 and 4-4). These results clearly suggest that artificial selection might have contributed to the domestication of *GSE9* in the *japonica* population, which further facilitated the divergence between *indica* and *japonica*. We have added these results and related information in this revised manuscript (lines 250-260 and 322-326). Fig. 5d was also modified in the revised version.

Minor comments.

Q4. The outgroup species of *japonica* and *indica* is likely to be *Oryza rufipogon* (Or-III) and *Oryza rufipogon* (Or-I), respectively. However, it is hard to identify the relationship from Fig. 5e. Please change the colors to distinguish *indica*, *japonica*, Or-

I, Or-II and Or-III in Fig. 5e easily. Furthermore, the relationship was addressed in some of previous analyses (Huang et al., *Nature* 2012). Therefore, please cite such the previous papers to enhance the evolutionary relationship.

Response: In this revised manuscript, we have sampled more *O. rufipogon* accessions to construct the phylogenetic tree, and used more distinguishable colors to show the branches of *indica*, *japonica*, Or-I, Or-II and Or-III. In addition, we have added one sentence and cited the reference (Huang et al., *Nature*, 2012, 490:497-501) to indicate that *indica* and *japonica* have been suggested to be descended from Or-I and Or-III, respectively (lines 269-270).

Q5. Although the authors concluded “Remarkably, none of the plant *de novo* genes have been shown the roles in phenotypic variation and morphological differentiation”, several papers introduced *de-novo* genes in plants. Please cite these papers as follows.

1. Takeda et al. A *de novo* gene originating from the mitochondria controls floral transition in *Arabidopsis thaliana*. Takeda et al., *Plant Molecular Biology* 2022.
2. Mingsheng et al., *QQS* orphan gene and its interactor NF-YC4 reduce susceptibility to pathogens and pests. 2018.

Response: Thank you for the above comments. We have changed this sentence into the following: “Most insights into *de novo* evolved genes come from large-scale comparative genome, transcriptome and proteome analyses thus far. Although the putative functions of several *de novo* genes have been reported in certain eukaryotic species, including *Drosophila* and *Arabidopsis*, the potential role of *de novo* genes remains unexplored in staple crops. None of these genes has been investigated in detail for their roles in morphological differentiation between subspecies.” (lines 313-318). The two references (Takeda et al., *Plant Mol Biol*, 2023, 111:189-203; Qi et al., *Plant Biotechnol J*, 2019, 17(1):252-263) have also been cited in the revised version (line 316).

Reviewers' Comments:

Reviewer #1:

Remarks to the Author:

The authors have answered all the questions I raised, and I think this manuscript can be accepted now.

Reviewer #2:

Remarks to the Author:

The authors have improved the previous version significantly, and I am satisfied with all their responses to my concerns. Additional well-conducted analyses have been added, the paper is better explained and quite readable. Overall, this is now an exceptional piece of work.

I have just minor points that the authors may wish to consider.

Regarding the Ka/Ks analysis that I suggested, the interpretation in this case appears more complex than I imagined. In particular, when we compare a GSE9-sequence to a gse9-sequence and obtain, e.g. a Ka/Ks ratio < 1 (excess of synonymous substitutions indicating purifying selection), it is not clear which of the two lineages actually accumulated those substitutions. For this reason, the gse9 and GSE9 types should be compared to an outgroup, which can only be *O. barthii*, *longistaminata* or *glumaepatula*. This has been done only partially. What ratios are observed when GSE9-japonicas and gse9-indicas are compared to one of these outgroups? I don't have any practical experience with this, but can the ratio be obtained on a population level (not on selected individuals)?

Finally, I suggest to replace 'any' with 'related' on line 302 (you would have to check all taxa outside *Oryza* to make that claim).

Peter Civan, 21/07/2023

Reviewer #3:

Remarks to the Author:

The authors have satisfactorily replied to the comments and questions from my first review.

Reviewer #4:

Remarks to the Author:

I read the comments about my queries. I then enjoyed reading the revised version. I'm fully satisfied with the revision. I think that this manuscript is deserved for a publication in Nature Communications.

Responses to comments by Reviewer #1

The authors have answered all the questions I raised, and I think this manuscript can be accepted now.

Response: Thank you very much for reviewing our manuscript and supporting its publication.

Responses to comments by Reviewer #2

Q1. The authors have improved the previous version significantly, and I am satisfied with all their responses to my concerns. Additional well-conducted analyses have been added, the paper is better explained and quite readable. Overall, this is now an exceptional piece of work.

I have just minor points that the authors may wish to consider.

Regarding the Ka/Ks analysis that I suggested, the interpretation in this case appears more complex than I imagined. In particular, when we compare a $GSE9$ -sequence to a $gse9$ -sequence and obtain, e.g. a Ka/Ks ratio <1 (excess of synonymous substitutions indicating purifying selection), it is not clear which of the two lineages actually accumulated those substitutions. For this reason, the $gse9$ and $GSE9$ types should be compared to an outgroup, which can only be *O. barthii*, *longistaminata* or *glumaepatula*. This has been done only partially. What ratios are observed when $GSE9-japonicas$ and $gse9-indicas$ are compared to one of these outgroups? I don't have any practical experience with this, but can the ratio be obtained on a population level (not on selected individuals)?

Response: The authors are truly grateful to this reviewer for the detailed and insightful suggestions and comments to improve our manuscript.

Following the reviewer's above suggestion, we used the improved branch-site model to detect positive selection that affects only a few sites on $GSE9-japonicas$ or $gse9-indicas$ at population level. The homologous sequences of 16 *O. barthii*, 17 *O. longistaminata* and 20 *O. glumaepatula* accessions were obtained from OryzaGenome, and then were used as background branches in the program *codeml* of PAML v4.9,

respectively. The sequences of *GSE9-japonicas* and *gse9-indicas* were used as foreground branches, respectively. A total of eight comparisons were performed to detect positive selection in foreground branches. Among them, the alternative hypothesis (free $d_N / d_S = 1$ or ω for foreground branches) of two comparisons, including *GSE9-japonicas* vs *O. barthii* and *GSE9-japonicas* vs mixed, suggested that positive selection has acted on the start codon in *GSE9-japonicas* with > 90% posterior probabilities of $d_N / d_S > 1$ (Response Table 1). However, neither of them rejected the null hypothesis (fixed $d_N / d_S = 1$ for the foreground branch) through the likelihood ratio tests (LRTs) at 0.05 level. The insignificance of the LRTs might be due to few mutation sites, and only 3 mutation sites with extremely low frequency were found in the population of *GSE9-japonicas*. Considering that LRTs cannot fully support positive selection in *GSE9-japonicas*, we chose not to add these results into the revised manuscript.

Response Table 1. Summary of statistics for detecting selective constrains on the reading frames of *GSE9* and *gse9* in *Oryza* species using the branch-site model in PAML.

Comparison	Foreground branches	Background branches	Hypothesis	LnL	Parameters	Positively selected sites (BEB, Pr > 0.9)
1	gse9-indicas	O. barthii	null	-513.4222	$p_0 = 0.5614, p_1 = 0.4386, p_{2a} = 0.0000, p_{2b} = 0.0000$ Background: $\omega_0 = 0.0000, \omega_1 = 1.0000, \omega_{2a} = 0.0000, \omega_{2b} = 1.0000$ Foreground: $\omega_0 = 0.0000, \omega_1 = 1.0000, \omega_{2a} = 1.0000, \omega_{2b} = 1.0000$	-
			alternative	-513.4222	$p_0 = 0.5614, p_1 = 0.4386, p_{2a} = 0.0000, p_{2b} = 0.0000$ Background: $\omega_0 = 0.0000, \omega_1 = 1.0000, \omega_{2a} = 0.0000, \omega_{2b} = 1.0000$ Foreground: $\omega_0 = 0.0000, \omega_1 = 1.0000, \omega_{2a} = 1.0000, \omega_{2b} = 1.0000$	NAN
2	gse9-indicas	O. longistaminata	null	-772.3650	$p_0 = 0.5872, p_1 = 0.4128, p_{2a} = 0.0000, p_{2b} = 0.0000$ Background: $\omega_0 = 0.0000, \omega_1 = 1.0000, \omega_{2a} = 0.0000, \omega_{2b} = 1.0000$ Foreground: $\omega_0 = 0.0000, \omega_1 = 1.0000, \omega_{2a} = 1.0000, \omega_{2b} = 1.0000$	-
			alternative	-772.3650	$p_0 = 0.5872, p_1 = 0.4128, p_{2a} = 0.0000, p_{2b} = 0.0000$ Background: $\omega_0 = 0.0000, \omega_1 = 1.0000, \omega_{2a} = 0.0000, \omega_{2b} = 1.0000$ Foreground: $\omega_0 = 0.0000, \omega_1 = 1.0000, \omega_{2a} = 1.0000, \omega_{2b} = 1.0000$	NAN
3	gse9-indicas	O. glumaepatula	null	-517.2728	$p_0 = 0.3802, p_1 = 0.4230, p_{2a} = 0.0932, p_{2b} = 0.1037$ Background: $\omega_0 = 0.0000, \omega_1 = 1.0000, \omega_{2a} = 0.0000, \omega_{2b} = 1.0000$ Foreground: $\omega_0 = 0.0000, \omega_1 = 1.0000, \omega_{2a} = 1.0000, \omega_{2b} = 1.0000$	-
			alternative	-517.2729	$p_0 = 0.2010, p_1 = 0.2236, p_{2a} = 0.2723, p_{2b} = 0.3031$ Background: $\omega_0 = 0.0000, \omega_1 = 1.0000, \omega_{2a} = 0.0000, \omega_{2b} = 1.0000$ Foreground: $\omega_0 = 0.0000, \omega_1 = 1.0000, \omega_{2a} = 1.0000, \omega_{2b} = 1.0000$	NAN

4	gse9-indicas	mixed	null	-894.5610	$p_0 = 0.5833, p_1 = 0.4167, p_{2a} = 0.0000, p_{2b} = 0.0000$ Background: $\omega_0 = 0.0000, \omega_1 = 1.0000, \omega_{2a} = 0.0000, \omega_{2b} = 1.0000$ Foreground: $\omega_0 = 0.0000, \omega_1 = 1.0000, \omega_{2a} = 1.0000, \omega_{2b} = 1.0000$	-
			alternative	-894.5610	$p_0 = 0.5833, p_1 = 0.4167, p_{2a} = 0.0000, p_{2b} = 0.0000$ Background: $\omega_0 = 0.0000, \omega_1 = 1.0000, \omega_{2a} = 0.0000, \omega_{2b} = 1.0000$ Foreground: $\omega_0 = 0.0000, \omega_1 = 1.0000, \omega_{2a} = 1.0000, \omega_{2b} = 1.0000$	NAN
5	GSE9-japonicas	O. barthii	null	-514.3225	$p_0 = 0.0000, p_1 = 0.0000, p_{2a} = 1.0000, p_{2b} = 0.0000$ Background: $\omega_0 = 0.0207, \omega_1 = 1.0000, \omega_{2a} = 0.0207, \omega_{2b} = 1.0000$ Foreground: $\omega_0 = 0.0207, \omega_1 = 1.0000, \omega_{2a} = 1.0000, \omega_{2b} = 1.0000$	-
			alternative	-513.3711	$p_0 = 0.0000, p_1 = 0.0000, p_{2a} = 1.0000, p_{2b} = 0.0000$ Background: $\omega_0 = 0.0207, \omega_1 = 1.0000, \omega_{2a} = 0.0207, \omega_{2b} = 1.0000$ Foreground: $\omega_0 = 0.0207, \omega_1 = 1.0000, \omega_{2a} = 999.0000, \omega_{2b} = 999.0000$	1 V 0.982*
6	GSE9-japonicas	O. longistaminata	null	-766.5964	$p_0 = 0.4955, p_1 = 0.3949, p_{2a} = 0.0610, p_{2b} = 0.0486$ Background: $\omega_0 = 0.0000, \omega_1 = 1.0000, \omega_{2a} = 0.0000, \omega_{2b} = 1.0000$ Foreground: $\omega_0 = 0.0000, \omega_1 = 1.0000, \omega_{2a} = 1.0000, \omega_{2b} = 1.0000$	-
			alternative	-766.5961	$p_0 = 0.4955, p_1 = 0.3949, p_{2a} = 0.0610, p_{2b} = 0.0486$ Background: $\omega_0 = 0.0000, \omega_1 = 1.0000, \omega_{2a} = 0.0000, \omega_{2b} = 1.0000$ Foreground: $\omega_0 = 0.0000, \omega_1 = 1.0000, \omega_{2a} = 1.0000, \omega_{2b} = 1.0000$	NAN
7	GSE9-japonicas	O. glumaepatula	null	-509.1890	$p_0 = 0.9999, p_1 = 0.0000, p_{2a} = 0.0000, p_{2b} = 0.0000$ Background: $\omega_0 = 0.5936, \omega_1 = 1.0000, \omega_{2a} = 0.5936, \omega_{2b} = 1.0000$ Foreground: $\omega_0 = 0.5936, \omega_1 = 1.0000, \omega_{2a} = 1.0000, \omega_{2b} = 1.0000$	-
			alternative	-509.1890	$p_0 = 1.0000, p_1 = 0.0000, p_{2a} = 0.0000, p_{2b} = 0.0000$ Background: $\omega_0 = 0.5936, \omega_1 = 1.0000, \omega_{2a} = 0.5936, \omega_{2b} = 1.0000$ Foreground: $\omega_0 = 0.5936, \omega_1 = 1.0000, \omega_{2a} = 2.5413, \omega_{2b} = 2.5413$	NAN

8	GSE9-japonicas	mixed	null	-894.0899	$p_0 = 0.5020, p_1 = 0.0518, p_{2a} = 0.4045, p_{2b} = 0.0417$ Background: $\omega_0 = 0.0494, \omega_1 = 1.0000, \omega_{2a} = 0.0494, \omega_{2b} = 1.0000$ Foreground: $\omega_0 = 0.0000, \omega_1 = 1.0000, \omega_{2a} = 1.0000, \omega_{2b} = 1.0000$	-
			alternative	-894.1295	$p_0 = 0.8655, p_1 = 0.0901, p_{2a} = 0.0403, p_{2b} = 0.0042$ Background: $\omega_0 = 0.0491, \omega_1 = 1.0000, \omega_{2a} = 0.0491, \omega_{2b} = 1.0000$ Foreground: $\omega_0 = 0.0491, \omega_1 = 1.0000, \omega_{2a} = 17.2895, \omega_{2b} = 17.2895$	1 V 0.918

Note: The likelihood ratio tests (LRTs) were performed to compare the null hypothesis with the alternative hypothesis. For one LRT, twice of the log likelihood difference between two models was compared against the chi-squared (χ^2) statistics, and the degree of freedom is one for alternative/null tests ($\chi_{0.05,1}^2 = 3.84$). BEB, Bayes Empirical Bayes analysis. Pr, posterior probability.

Numbers indicate the amino acid position in *japonica* and its posterior probability of positive selection.

Q2. Finally, I suggest to replace 'any' with 'related' on line 302 (you would have to check all taxa outside *Oryza* to make that claim).

Response: Many thanks for this suggestion. We have modified the sentence into the following: “Here, we found that *GSE9* is an *Oryza*-specific gene that is absent in related taxa outside the genus *Oryza*.” (lines 301-302).

Responses to comments by Reviewer #3

The authors have satisfactorily replied to the comments and questions from my first review.

Response: We are pleased that our revisions were able to address all of your comments and questions to your satisfaction.

Responses to comments by Reviewer #4

I read the comments about my queries. I then enjoyed reading the revised version. I'm fully satisfied with the revision. I think that this manuscript is deserved for a publication in *Nature Communications*.

Response: The authors sincerely thank the reviewer for the positive evaluation of our work. We are pleased that our revisions were able to address all of your suggestions and comments.

Reviewers' Comments:

Reviewer #2:

Remarks to the Author:

I am satisfied with the authors' response and with the current version of the manuscript.

Responses to comments by Reviewer #2

I am satisfied with the authors' response and with the current version of the manuscript.

Response: Thank you again for efforts putting in our manuscript. We are pleased that our revisions were able to address all of your comments to your satisfaction.